# Temporal recurrence as a general mechanism to explain neural responses in the auditory system
Ulysse Rançon [1] ✉, Timothée Masquelier[1,3] & Benoit R. Cottereau [1,2,3]

Computational models of neural processing in the auditory cortex usually ignore that neurons have an internal memory: they characterize their responses from simple convolutions with a finite temporal window. To circumvent this limitation, we propose here a new, simple and fully recurrent neural network (RNN) architecture incorporating cutting-edge computational blocks from the deep learning community and constituting the first attempt to model auditory responses with deep RNNs. We evaluated the ability of this approach to fit neural responses from 8 publicly available datasets, spanning 3 animal species and 6 auditory brain areas, representing the largest compilation of this kind. Our recurrent models significantly outperform previous methods and a new Transformer-based architecture of our design on this task, suggesting that temporal recurrence is the key to explain auditory responses. Finally, we developed a novel interpretation technique to reverse-engineer any pretrained model, regardless of its stateful or stateless nature. Largely inspired by works from explainable artificial intelligence (xAI) –a field focused on enhancing the interpretability and transparency of complex machine learning models–, our method suggests that auditory neurons have longer memory (several seconds) than indicated by current techniques based on the Spectro-Temporal Receptive Field (STRF). Together, these results highly motivate the use of deep RNNs within computational models of sensory neurons, as protean building blocks capable of assuming any function.

The activity of a neuron at a given time instant not only depends on its current synaptic inputs but also on its past ones, as well as on the history of its own state. This property is widely taken into account in foundation theories of neural computation[1–3] and extensively observed in the sensory systems of numerous animal species and notably along the auditory pathway. A range of adaptive neural processes have been described in this context, including mechanisms such as stimulus-specific adaptation (SSA)[4,5] and short-term plasticity (STP)[6,7], as well as functional phenomena like contrast gain control (CGC)[8–10], context dependence[11,12] or priming[13,14]. The latter were described by experimentalists over the last two decades and collectively illustrate how sound processing is dynamically modulated by recent stimulus history and internal neural states. Although it is now widely accepted that sound processing in biological systems is constantly modulated by these representations[15,16], the most common computational models of neural processing in the auditory system do not fully incorporate them. Indeed, these models are based on spectro-temporal receptive fields (STRFs,[17,18]) and consist in a cascade of convolutions. Despite their

simplicity of implementation and interpretation, they face several issues for fitting neural responses. First of all, their temporal window duration is chosen manually, based on previous knowledge of integration timescales in the targeted area and neuron class for the given task. As a result, most models have their STRFs restricted to durations of a few hundred milliseconds[6,19–22], making them incapable of modulating their responses over larger timescales as highlighted in several electrophysiological studies[5,11,23]. Secondly, the finite duration of this window creates a sharp boundary between the part of the auditory stimulus that contributes to the response and the one that does not, which is not biologically plausible. Finally, several studies showed that STRF-based models of neural responses in auditory cortex should integrate stimulus information over large temporal windows (hundreds of ms) to improve their neural fitting performances[24], although such delays are unrealistic in the nervous system[25].

As a partial fix to these issues, conventional models have been extended with nonlinear recurrences akin to the observations mentioned above (SSA,

[1]Univ Toulouse, CNRS, CerCo, Toulouse, France. [2]IPAL, CNRS IRL 2955, Singapore, Singapore. [3]These authors contributed equally: Timothée Masquelier, Benoit R. Cottereau. ✉e-mail: ulysse.rancon@gmail.com

STP, CGC,...), which successfully boost their memory capacity and performances[6,7,26,27], while only requiring a narrow range of delays[25]. These operations are nonetheless only monofunctional add-ons to existing stateless backbones. They still rely on spectro-temporal convolutions and thus keep the problems related to the fixed duration of the temporal window used to process the auditory inputs. Furthermore, most of these studies proposed a different form of recurrence and never attempted to combine them into a single and unified framework, as if they were incompatible, although properties such as STP, SSA, or CGC (among others) could be the manifestation of the same and more general computation principle (see e.g.,[28] or ref. [29]). The manual derivation of the nonlinearities is yet another obstacle to get closer to the underlying mechanism governing neural processing and allowing to fit sensory responses better[30,31].

To fully address these limitations, we propose here a new class of computational models based on deep recurrent neural networks (RNNs) to characterize sound processing in the auditory pathway. We name these models StateNets because they are stateful in time and can potentially internalize any temporal dynamics relevant to reproduce sensory neural behavior[32–34]. In line with the observations reported above, their main principle is that the current activity of a neuron relies on an internal and high-dimensional hidden state whose current value depends on previous states as well as on the history of auditory inputs[35]. To the best of our knowledge, this constitutes the first attempt to model auditory responses with deep RNNs, although recurrent mechanisms have been extensively associated with local microcircuits[15,36] and that audition is a very temporal modality. Using electrophysiological data collected in a wide range of animal species and brain areas, we show that StateNet models outperform stateless STRF-based models at fitting auditory neural responses, therefore offering a drastic improvement over the state-of-the-art. In addition, their design is free from unrealistically long synaptic delays and allows to get rid of the crucial hyperparameter that is still manually defined in current modeling studies: the temporal receptive fields (TRFs) duration. Furthermore, the differentiability of our models, combined with recent advances in the interpretability of deep learning architectures—a field referred to as explainable Artificial Intelligence (xAI), see Gilpin et al.[37]—allowed us to attribute these increased performances to Long Short-Term Memory (LSTM) capacity. We propose here a new method inspired from this literature[38–40] to reveal and interpret STRF-like features for such time-recurrent models, thereby circumventing the lack of explicit spatial-temporal weighting. This method was crafted to closely mimic experimental practices[17,41,42] and generalizes the linear STRF for any model, without any constraint. Our validation benchmark constitutes a precious compilation in the field of auditory neural response modeling, with 11 models (CNNs, a Transformer, and RNNs) tested on 8 datasets, recorded from 3 animal species (ferret, rat, zebra finch) and 5 auditory brain areas (MGB, A1, PEG, MLd, Field L). To foster comparative studies between models, our codes are made publicly available on our online repository (https://github.com/urancon/deepSTRF). Overall, our study strongly supports the notion that temporal recurrence is a general feature of sensory neurons. It also provides additional demonstration that combining computational neuroscience and AI opens interesting perspectives for better understanding computations in the nervous system.

## Results

This work introduces a new class of models—StateNet—based on deep recurrent networks to explain neural responses in the auditory system. As observed in electrophysiological recordings, these models possess an internal memory and are sensitive to long-term dependencies in their inputs. In the following, we first demonstrate that the performance of StateNet models in neural response fitting tasks is significantly higher than that achieved with state-of-the-art models of auditory processing and recent attention-based models (i.e., Transformers). Next, we show that they only require a single timestep to provide accurate temporal predictions, addressing the issue of implausible delays faced by previous models. Finally, we propose a new framework derived from recent developments in xAI that

permits to determine the auditory properties that maximize neural responses in StateNet models.

In the following of this article, we use the terms "RNN" and "stateful models" to designate DNet and StateNet models, in opposition to the remaining models, qualified as "stateless". Among StateNet models, the GRU, LSTM, and Mamba versions are labeled as "Gated RNNs" because of their corresponding recurrent blocks.

### Recurrent networks outperform state-of-the-art models and Transformers on a large gamut of datasets

We trained a series of neural network-based models to predict the neural activity of auditory cortical single units, given sound stimuli in the form of spectrograms. We used eight different datasets (AA1 MLd, AA1 Field L, NAT4 PEG, NAT A1, NS1, Wehr, Asari MGB, and Asari A1) that combined neural recordings (extra and intracellular) performed in a large panel of auditory areas and species (rat, ferret and zebra finch)[22,31,43,44]—see section "Electrophysiology datasets of auditory responses". As baseline models, we chose a series of popular backbones, based on the Linear (Nonlinear) STRFs model[17,45,46], as well as a 2D Convolutional Neural Network (CNN) proposed in a recent study and that processes the spectrogram in a similar manner as an image[31]. Among these models, DNet[25] is a recurrent version of the NRF model[22]: its hidden units are stateful with leaky dynamics akin to a non-spiking Leaky Integrate-and-Fire unit[2]. We also developed a Transformer-based model[47] as well as an RNN backbone, thereafter referred to as StateNet with an interchangeable stateful core. StateNet models included different recurrent architectures: Elman RNNs, LSTM, GRU, S4 and Mamba[48–52]. Model performances ($CC_{norm}$) are reported in Table 1.

Stateless models were tested using different temporal integration window durations. The best performances across durations are reported here. On average (see the last column in Table 1), except for the simple Elman RNN, all StateNet models significantly outperform the previous state-of-the-art, which is given here by the 2D-CNN models (in yellow, see Pennington and David[31]). The GRU model notably permits to reach an average $CC_{norm}$ value of 51.2%, which improves by about 10% the 2D-CNN average score (46.7%). Interestingly, these StateNet models also outperforms Transformers at this task, even though the latter already offers improvement over the previous state-of-the-art on average $CC_{norm}$ (47.4%). Among StateNet models, the best performances are obtained with the GRU, followed by Mamba, S4, LSTM, and finally Elman RNN models. To go further into details, across all datasets, StateNet models are always among the two best ones and lead to the best performances in all datasets except three (NAT4 PEG and Asari MGB and A1). For NAT4 PEG, the LSTM, GRU, S4, and Mamba models outperform the previous state-of-the-art (~54.6% on average across the 4 models versus 50.2% for DNet) but provide a lower score than our newly proposed Transformer. For Asari MGB and A1, the best-performing model is DNet, which is also recurrent, closely followed by StateNets. To better illustrate the neural fitting performances of our models, we show a few examples of predictions and ground-truth recordings from well-predicted neurons in different datasets in Fig. 1.

### Recurrent networks only need a few timesteps for neural data prediction

Recurrent models (StateNet but also DNet) build their predictions of neural activity at the current time step from a very short temporal window of past stimulus inputs (e.g., 5 timesteps for DNet and as short as 1 timestep for StateNet models). In preliminary experiments, we tried feeding a longer stimulus history to these models, but this did not result in better performances. This is consistent with observations by ref. [25] and can be explained simply by the fact that RNNs are already well-equipped to process sequential data. Therefore, contrary to previous stateless approaches (L, LN, NRF, 2D-CNN) and to Transformers, these models do not need to process auditory stimuli across time windows of arbitrary fixed time durations. To illustrate how performances scale with the duration of the time window that is accessible, we show in Fig. 2 the performances of the different models as a function of the temporal context. In all datasets (panels a–h), StateNet

**Table 1 | Neural response fitting performances of the 11 tested models for 8 different datasets**

| Model | Dataset | CC$_{norm}$ [%] | | | | | | | | Mean CC$_{norm}$ |
|---|---|---|---|---|---|---|---|---|---|---|
| | | AA1 | | NAT4 | | NS1 | Wehr | | Asari | |
| | | MLd | Field L | PEG | A1 | | | MGB | A1 | |
| | L | 56.4 | 53.2 | 37.5 | 44.0 | 54.0 | 21.0 | 16.1 | 17.5 | 37.3 |
| | LN | 59.0 | 52.9 | 40.4 | 48.4 | 56.7 | 17.9 | 17.2 | 16.6 | 38.6 |
| | NRF | 61.2 | 56.3 | 48.9 | 59.8 | 63.3 | 23.6 | 17.8 | 19.0 | 43.8 |
| | DNet | 60.1 | 61.3 | 50.2 | 59.3 | 58.9 | 24.3 | **22.0** | **20.2** | 44.5 |
| | 2D CNN | 68.9 | 65.0 | 48.0 | 62.0 | 70.1 | 22.5 | 19.5 | 17.6 | 46.7 |
| | Transformer | 68.3 | 65.5 | **55.5** | 64.4 | 73.0 | 18.0 | 19.5 | 15.1 | 47.4 |
| StateNet | Elman RNN | 61.5 | 54.4 | 50.1 | 60.2 | 64.6 | 16.8 | 18.0 | 10.6 | 42.0 |
| | GRU | <u>73.0</u> | **71.0** | 54.1 | <u>64.8</u> | **75.1** | <u>31.0</u> | <u>20.6</u> | <u>19.6</u> | **51.2** |
| | LSTM | 71.4 | 68.6 | <u>54.8</u> | **65.1** | <u>74.3</u> | 23.0 | 19.8 | 16.7 | 49.2 |
| | S4 | 70.8 | <u>70.6</u> | 54.7 | 64.4 | 71.4 | **31.2** | 14.7 | 16.9 | 49.3 |
| | Mamba | **73.4** | 69.9 | 54.7 | 64.2 | 71.4 | 30.5 | 17.2 | 15.4 | <u>49.6</u> |

Average performances across datasets are provided in the last column. Previous and new (Transformer, StateNet) model classes are respectively shown in the upper and lower rows of the table. Performances are given as the noise-corrected Pearson correlation coefficient, in %. For stateless models, the highest value across the different time windows tested is provided (see Fig. 2). For each dataset and model, CC$_{norm}$ values were averaged across neurons and training seeds. **Bold** and <u>underlined</u> fonts, respectively, indicate the best and second-best models on a given dataset.

models are ideally positioned at the upper-left corners of the plots and thus provide an ideal trade-off between performances and temporal context duration. Other models generally increase their performances for longer temporal windows, even though saturation and overfitting effects can be observed in some datasets (e.g., AA1 MLd and Field L). These observations are in accordance with prior published work[11,25]. Interestingly, the DNet model systematically outperforms its stateless counterpart (NRF model), no matter the length of the input temporal window, and despite an almost equal parameter count. It illustrates the computational advantage brought by implicit recurrence. Furthermore, by learning the adequate temporal integration timescales to fit the data, StateNet models do not depend on the temporal window size hyperparameter, which is otherwise often manually defined to implausibly long delays[25].

### A new method to characterize the selectivity of auditory neurons in recurrent networks

Auditory selectivity in neural units can be interpreted through the STRFs of stateless models[17,18]. They represent the weights assigned to each frequency-time bin in the construction of the response and can be estimated using reverse correlation techniques based on synthetic stimuli (usually white noise, see De Boer and Kuyper[41] and Aertsen and Johannesma[17]) but also natural stimulus ensembles (see Theunissen et al.[18,19]). However, this approach is not directly applicable to StateNet models, as they explicitly weight only spectral information while their temporal integration remains implicit. Therefore, we propose a new technique (″GradMaps″ and ″Dreams″) directly inspired from advanced xAI methods (″Deep Dream″, see Olah et al.[39], but also Selvaraju et al.[40]) and which allows to compute the preferred stimulus of a target unit in any differentiable model (see the "Model interpretability with feature visualization: gradient maps and deep dreams" section in "Materials and Methods"). A graphic illustration of the approach is shown in Fig. 3a. Starting from a null spectrogram ($x_0$), a forward pass (indicated by red arrows) through the trained and frozen network yields predicted responses (which are to be maximized) and the associated loss. Backpropagation (blue arrows) produces a gradient map ($g_0$) which is then subtracted from the initial spectrogram ($x_0$) to generate a new stimulus ($x_1$). This new stimulus elicits stronger responses in the neural unit under consideration. This process is repeated until an early stopping criterion is reached, iteratively creating spectro-temporal inputs (Dreams) that maximize the neuron's response.

Figure 3b shows examples of gradient maps (GradMaps) and Dreams obtained at the initialization and at the end of the process for a single unit of

the NS1 dataset using different models. (NRF, DNet, RNN, GRU and Transformer). GradMaps obtained from both stateless (NRF) and stateful (DNet, RNN and GRU) models after the first iteration ($g_0$) exhibit a similar structure, suggesting that they all capture the same functional property of the neuron to which they have been fitted. In this particular example, they show that the latter tends to be excited (inhibited) by sounds around 8 Hz (15 Hz) presented 50 ms ago. We also observe that GradMaps of recurrent models are smooth, especially along the temporal axis. This smoothness can be explained by the autoregressive nature of the response, where each time step is conditioned by the previous one. Similarly, the smoothness in the spatial/frequency domain for StateNets can be explained by the initial locally connected (LC) scheme applied in this dimension for information compression. Very importantly, GradMaps also resemble and include the STRF obtained from a Linear model (also shown on the figure for comparison).

Interestingly, despite their high level of nonlinearity and performance compared to conventional stateless models, the attention mechanism within Transformers leads to noisy GradMaps, making them more difficult to relate to STRFs. Their different aspect can be partly explained by the tokenization of the input spectrogram in frequency vectors of a single time step. Further investigation of this model should definitely look into how the shape of the tokens can improve the neural response fitting performances and the relatability of GradMaps to canonical STRFs obtained with STA. Other examples obtained from other models are provided in Supplementary Fig. S3.

In brief, GradMaps generalize the STRF and represent the linear portion of the neurons' input-output function. Alternatively, they can be viewed as the direction in which the stimulus should change in its own space to elicit stronger responses.

### Analysis of functional equivalence between models

To gain deeper insights into the functional distinctions between our proposed StateNet architectures and more conventional methods, we conducted an analysis of the similarity of GradMaps across various models. For each dataset, the GradMaps associated with each neural unit and architecture were flattened along the frequency and time dimensions. The similarity between the resulting vectors for each pair of models was then computed using Pearson's correlation coefficient. We show in Fig. 4a the similarity matrix obtained by averaging the correlation coefficients across all neurons in each dataset and then across datasets. This matrix reveals two distinct clusters: one comprising most STRF-based models (linear, LN, NRF and DNet) and the other consisting of StateNet models (outlined by blue

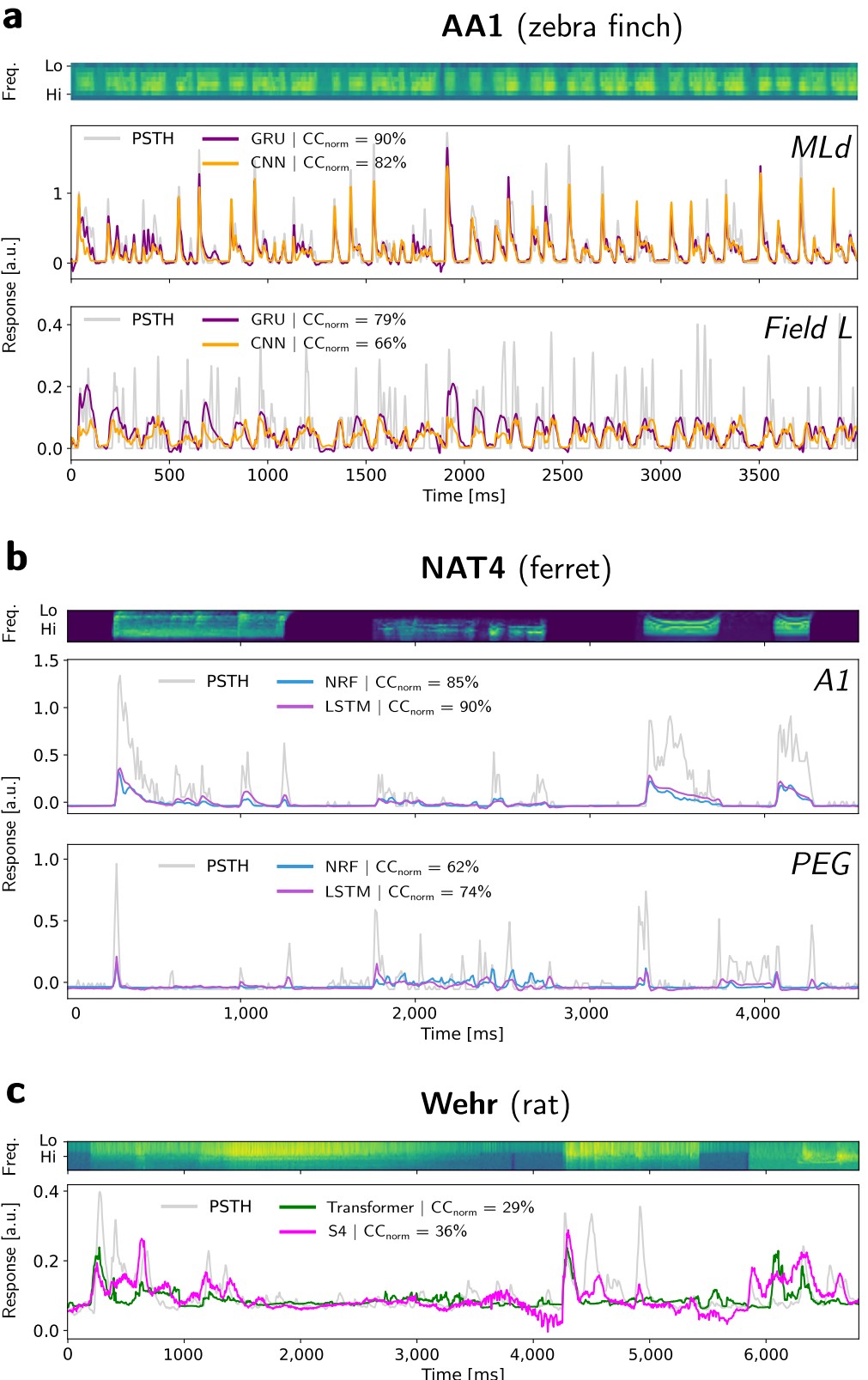

**Fig. 1 | Examples of model predictions vs actual neurophysiological recordings for different datasets.** Each panel shows the spectrogram of the stimulus (upper part) and the model predictions (in colors) of the peri-stimulus time histogram (PSTH, in gray) in normalized units (a.u.) as a function of time. $CC_{norm}$ values are computed over the entire test set of the shown neuron for each model. **a** Recordings from neurons in areas MLd and Field L (AA1 dataset, zebra finch). **b** Recordings from neurons in areas A1 and PEG (NAT4 dataset, ferret). **c** Recordings from a neuron in area A1 and PEG (Wehr dataset, ferret).

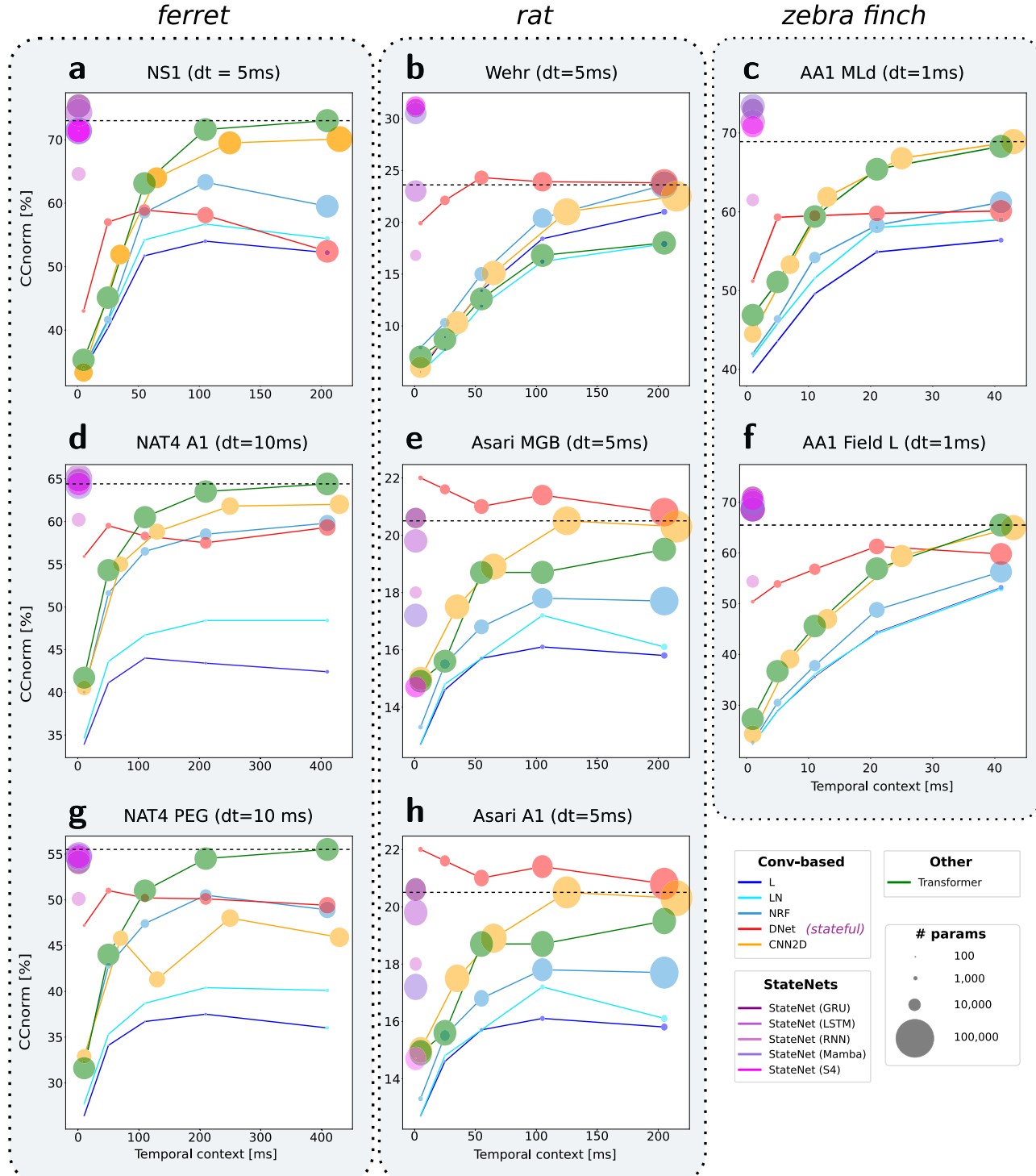

**Fig. 2 | Neural response fitting performances as a function of temporal context length.** The 11 models (colored disks) were tested on 8 different datasets (**a**–**h**) while varying the extent of their explicit temporal integration window (i.e., the duration of previous inputs they are provided to predict the responses in the current time-step). Because the size of conventional models varies with this STRF window, the number of learnable parameters is provided for reference as the disk diameter. Animal illustrations were AI-generated with DALL-E 3 (OpenAI). The diameters of the disks provide the number of parameters of the associated model. Best viewed in color on a monitor.

and purple boxes, respectively). It demonstrates that models within these two classes tend to generate similar GradMaps. Within the first cluster, similarity scores are higher between the linear and LN models, and between the NRF and DNet models. Interestingly, the GradMaps obtained from the Transformer and 2D-CNN models stand apart, further confirming observations made with the exemplar neurons shown in Fig. 3 and Fig. S3.

## Energy distribution in GradMaps correlates with model architecture

To go further, we now characterize how temporal integration differs between models, and notably between stateless and stateful architectures. We computed the GradMap energy across neurons and datasets for the clusters identified in Fig. 4a. These memory traces are shown in Fig. 4b. The

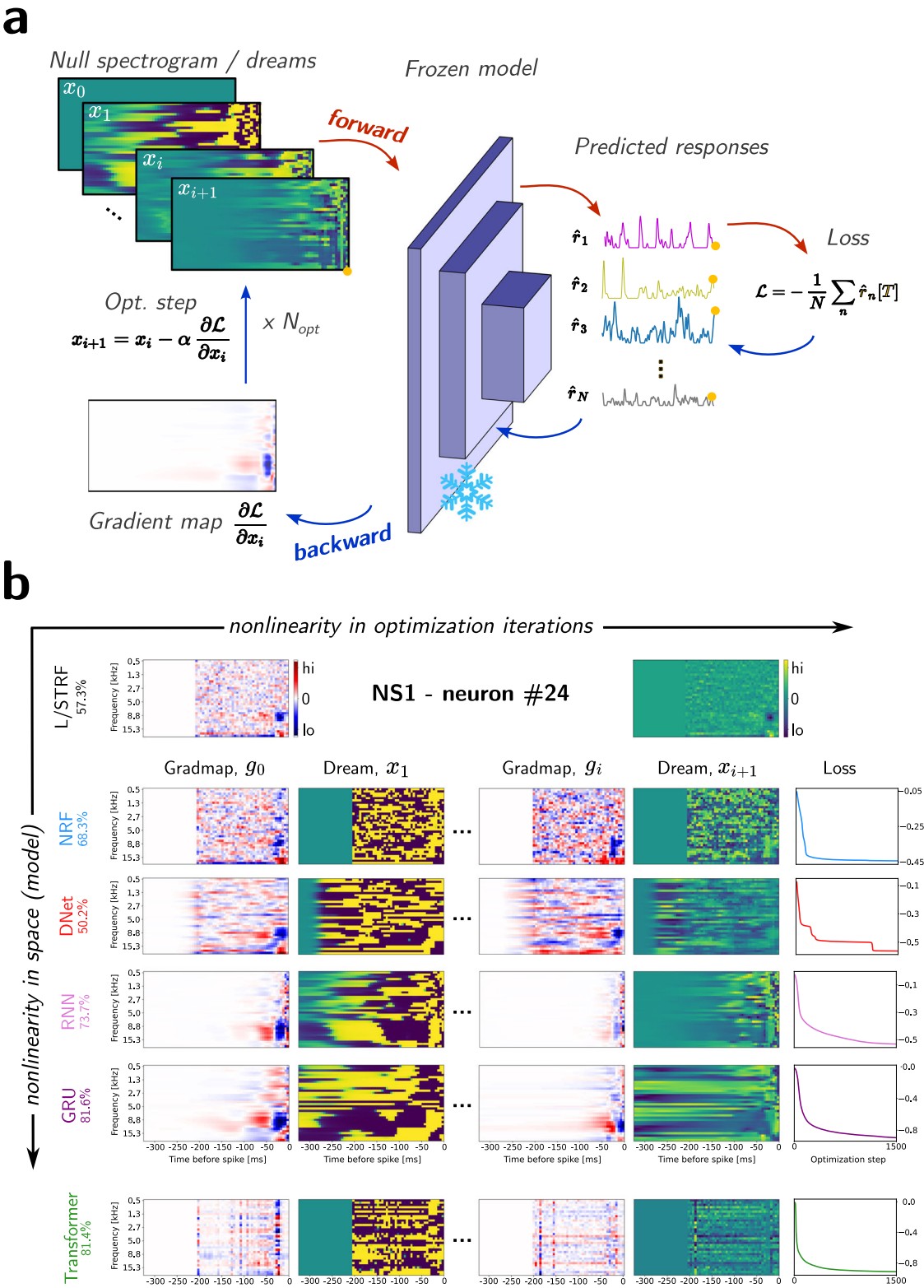

**Fig. 3 | Characterizing the selectivity and preferred stimuli of auditory neuron models with GradMaps. a** Illustration of the method. After learning, the parameters of the model are frozen. From a null spectrogram ($x_0$), a forward pass through the network (red arrows) provides the predicted responses and the associated loss. Then, a gradient descent (blue arrows) leads to a gradient map ($g_0$), which is subsequently subtracted from the initial spectrogram ($x_0$) to obtain a new stimulus ($x_1$) that triggers stronger responses in the considered neural unit. This process is repeated n times, leading to the spectro-temporal stimulus ($x_n$) that maximizes the neuron's response. **b** Illustration of the process on one neuron (#24) from the NS1

dataset for different models: NRF, DNet, RNN, GRU, and Transformer (CC$_{norm}$ values obtained with this neuron are displayed with the names of the models). For each model, the panel shows the first gradient map ($g_0$) and the subsequent Dream ($x_1$), the last gradient map ($g_n$) and its subsequent Dream ($x_{n+1}$), and the evolution of the loss function across iterations. For each Dream, we also provide the associated neural responses as a function of time. To improve the readability of the figure, we only show here the 300 last ms before the neuron discharge, but for some recurrent models, effects can be observed well before this limit. We provide in Supplementary Fig. S3 the full window of activation obtained with a GRU model for this neuron.

**Fig. 4 | GradMap characterization. a** Functional similarity matrix between models averaged across neurons and datasets. Each element represents the correlation coefficient between the GradMaps derived from two trained models of different architectures. Main clusters were highlighted in blue (STRF-based models) and purple (proposed State-Net). **b** GradMap energy as a function of latency, for different clusters of models and averaged across neurons and datasets. For easier comparison between models, traces were normalized by their maximum value.

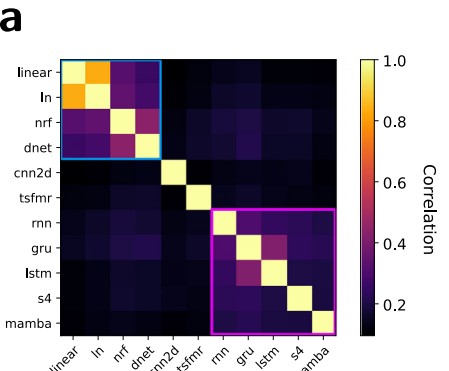

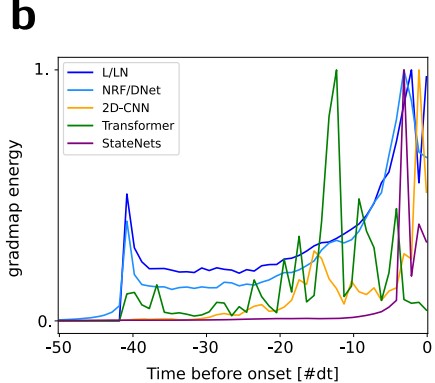

energy in GradMaps of StateNet models (in purple) does not appear to occupy longer durations compared to other models. Instead, it is concentrated at lower latencies, whereas stateless models maintain high levels of energy up to the limit of their STRF. This confirms that recurrent models primarily build their predictions of neural activity at the current time step from a very short temporal window of past stimulus inputs, as outlined in the results of Fig. 2. This observation is in agreement with recent electrophysiological studies in humans and ferrets, which have reported that auditory neurons in these species mostly rely on the recent past[53,54]. Nonetheless, it does not imply that StateNet models do not use older time steps as their neurons have an internal memory which keeps track of their previous states.

## Truncated back-propagation through time reveals the long-term memory of recurrent models

Unlike stateless models, recurrent models have the ability to integrate information from auditory stimuli of infinite duration. We examined the extent to which past auditory information influences the prediction of neural responses at the current time step. We trained our StateNet GRU architecture on the NS1 dataset using a modified version of the back-propagation through time (BPTT) algorithm, known as truncated back-propagation through time (TBPTT, see the corresponding "Materials and Methods" section). The selection of this dataset and model for our ablation study is justified by the lengthy sequences and their high temporal resolution (5 s divided into 1000 time steps of 5 ms each), as well as the very high level performance levels achieved (nearly 80% $CC_{norm}$, see Table 1).

First, we trained these new GRU models on the same random seeds as in Table 1 using TBPTT with warmup and a varying graph duration of $K = \{1, 10, 20, 50, 100, 500, 999\}$ time steps. The performances of our StateNet GRU model are shown in Fig. 5a; we can observe that they decrease monotonically with the length $K$ of the automatic differentiation graph. At high values of $K$, this decline in performance is relatively modest but remains significant. (e.g., 0.6% at $K = 500$ and 2.7% at $K = 100$), indicating that although there is limited information available from the distant past, the model successfully captures essential temporal dependencies required to predict current activity. Furthermore, it still outperforms the best stateless models from the literature (the 2D-CNN), even for $K$ values similar to the largest temporal integration window for this model. The most striking difference observed is at the extreme case $K = 1$, in which GRU and LSTMs reach test $CC_{norm}$, above 50%, which is far greater than conventional models with access to only one time step to compute gradients (<35%, see Fig. 2a). This further suggests that temporal recurrence serves as a more effective core computational principle for modeling auditory neural responses, operating robustly across both short and long time scales.

Next, to ensure a fairer comparison with stateless models, which do not have access to any prior information before the computational graph constructed by autograd, we trained an additional set of StateNet GRU models for the same values of $K$, but without warmup. Consequently, the graph is

constructed from the default null hidden state, rather than from a state refined through multiple warmup iterations. As in the previous experiment, the results are presented in Fig. 5a. As expected, performance decreased with $K$ in a manner similar to the previous scenario, but the decline was steeper, and performance remained consistently lower compared to the warmup condition. At $K = 1$, the performance of these models drops to levels comparable to those achieved by stateless models (around 30% $CC_{norm}$). Interestingly, this result demonstrates that RNNs, through warmup time steps, can leverage temporal dependencies extending beyond the length of the computational graph used during training. This in turn explains the very important dream duration observed for the most sophisticated StateNet models.

To further analyze how the training process of StateNet models influences their temporal processing capabilities, we computed the GradMaps of GRU models trained under standard conditions, with and without warmup in TBPTT, and examined their energy as a function of latency. Fig. 5b shows that for models trained with full BPTT instead of TBPTT, the GradMap energy is more distributed across time and less concentrated in recent time bins. A similar observation can be made between both TBPTT models. This strongly supports the idea that incorporating a warmup phase and a maximal graph length $K$ facilitates the learning of both long and short-term temporal dependencies. Moreover, they also alter the qualitative nature of temporal integration, as demonstrated by the additional GradMaps examples of these specific models shown in the Fig. 5c.

## Optimal Dream stimuli with long durations suggest new closed-loop experiments

Iteratively applying GradMaps to the null stimulus $x_0$ produces optimal stimuli (Dreams, $x_i$) eliciting high neural activity; this section aims to describe their properties.

Similar to the GradMaps to which they are closely related, the Dreams of StateNet models are inherently smooth and do not require explicit regularization, unlike stateless approaches. This smoothness emerges naturally from the temporal recurrence in the models, eliminating the need for careful hyperparameter tuning, such as L1 or L2 penalties. A convincing manifestation of this is the smoother Dream of DNet compared to the Dream of its stateless counterpart (NRF) (see Fig.3).

In the time domain, optimized Dreams of conventional stateless models and of the Transformer end abruptly at the limit of their STRF, while they gradually fade away in the case of recurrent models (DNet and State-Net). If the energy generally vanishes out at shorter latencies for some StateNet models (e.g., RNN), this is not true for all models (e.g., GRU, S4, Mamba), which exhibit non-zero values even in the earliest time steps. This observation supports the idea that these models possess extended short-term memory capacities.

To determine whether these initial Dream patterns indeed drive neural activations, we examined the contribution of each time step in the Dreams to

**a**

**b**

**c**

**Fig. 5 | Truncated backpropagation through time (TBPTT): results. a** StateNet performances as a function of TBPTT parameter *K*, in number of time steps (5 ms for NS1). The best stateless baselines are also shown for comparison: 2D-CNN (best in the previous literature), Transformer (best as of the present article). **b** GradMap memory traces for the StateNet GRU model on NS1, depending on the training procedure. Curves are averaged across output units (*N* = 73), and shaded areas correspond to ±one standard deviation. **c** GradMaps of each StateNet GRU model for neuron #65.

the maximization of the response. Again, these analyses were performed on the NS1 dataset for the same reasons as in the TBPTT subsection. Specifically, each StateNet model was frozen after training, and long Dreams of 1 up to *T* = 200 time steps (1 s) were optimized until loss convergence. These optimized Dreams were then input into their respective models. The resulting activations at the final (most recent) time step were averaged across neurons (*N* = 73) and models of the previously identified functional clusters (see Fig. 4a), and plotted against the Dream length in Fig. 6. As shown in the figure, the activation elicited by the Dreams increases almost monotonically with their length. The shape of the curves (steep initially and then saturating) indicates that the most recent time steps of the Dreams contribute more to the elicited activation than the earlier ones. The computationally weaker vanilla RNN model saturates rapidly, while the more powerful gated RNNs and SSM models continue to increase their activation for up to one full second without reaching saturation. We can therefore safely conclude that the long patterns present in the Dreams of our most powerful StateNet models are not optimization artifacts, but instead actively contribute to maximizing neural responses.

Finally, this ability to leverage long-range temporal dependencies is not present in randomly initialized networks, but rather inherited from training on the neural response fitting task (see Supplementary Fig. S3).

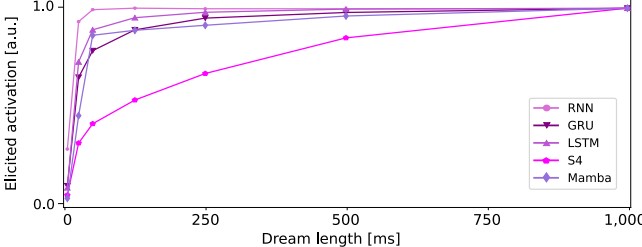

**Fig. 6 | Contribution of each Dream time step to the elicited activation.** Each curve was averaged across neurons (*N* = 73) and then normalized by its maximum to ease the comparison between models.

These results call for further validation in an experimental setting. Specifically, can longer stimuli (on the order of seconds) elicit stronger responses in a closed-loop system?

## Discussion

In this paper, we present a new class of models (*StateNet*, see Fig. 7c) based on deep recurrent networks to explain neural responses in the auditory

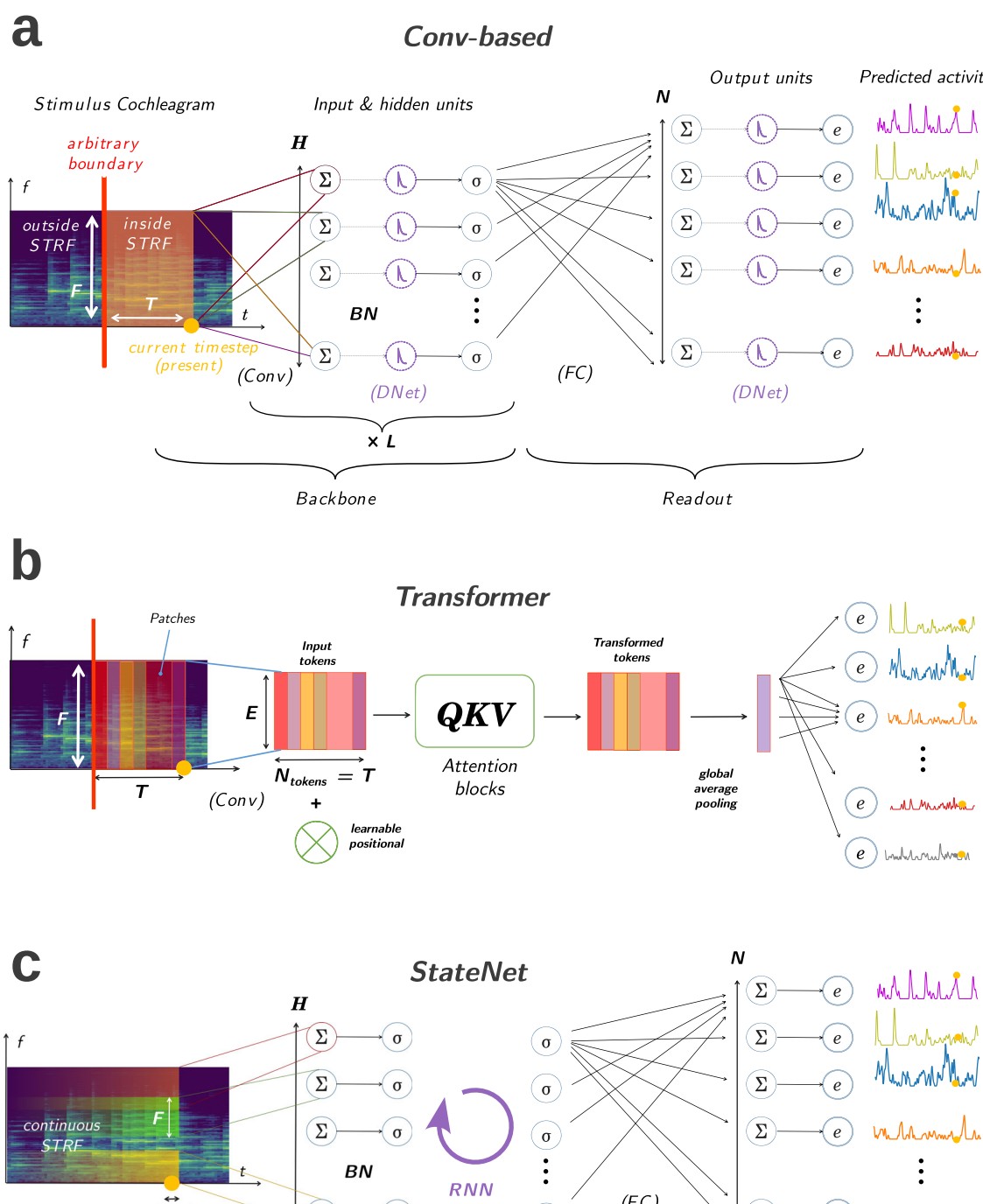

**Fig. 7 | Illustrated architectures computational models of auditory neural responses. a** General architecture of canonical conv-based models. The waveform-to-spectrogram transform is not represented here and assumed to have been applied prior to the model. A set of $H$ small or large spectro-temporal windows are convolved along the temporal dimension of the cochleagram and followed by some Batch Normalization (BN) and a sigmoid function as an input layer. This operation is repeated $L$ times and defines the backbone of the model: learnable parameters and computations that are common to all neuronal units to fit. From the embedding of the stimulus given by the backbone, the readout layer produces predictions of neural activity for each unit in the recorded population, with non-shared, specific parameters including weights, biases, and parametric double exponential output non-linearities. **b** Our proposed Transformer model. To give a prediction of current neural activity at time $t_0$, the network uses a temporal window of size $T$ of the previous stimulus spectrogram, similar to CNNs. This spectro-temporal window is decomposed into a number of non-overlapping patches, which are projected into an embedding space of defined dimension $E$ thanks to a 2d convolution. We call the vectors associated to each patch as a result of this operation *tokens*. In our proposed architecture, tokens are the projection of vertical, rectangular patches that are flat, spanning all frequencies but a single time-step. They are then fed to a Transformer block, notably composed of several layers of multi-head attention. The output of this block is an equal number of transformed tokens of the same dimensionality, which are averaged into one. From the latter, vector population activity at the current timestep is directly and linearly readout, with a learnable parametric double-exponential output activation function. **c** *StateNet* architecture. At each time-step, the current frequency vector is downsampled by a Locally Connected layer maintaining tonotopy, followed by BN. The resulting tonotopic projection is then given to the stateful bottleneck composed of a stateful/recurrent module, which updates its hidden state. Finally, a prediction of neural activity at the current timestep is performed as a linear projection from the latter.

system. To the best of our knowledge, we constituted the largest online repository for auditory neural response fitting, with 11 models benchmarked on 8 datasets (AA1 MLd, AA1 Field L, NAT4 PEG, NAT A1, NS1, Wehr, Asari MGB and Asari A1) recorded in large panels of auditory areas (MLd, Field L, PEG, MGB, and A1) and species (rat, zebra finch and ferret). We showed that most architectures of this model family (i.e., LSTM, GRU, S4 and Mamba) better fit neural responses than previous approaches proposed in the literature (see Table 1 and also Fig. 2). Notably, the Gated Recurrent Units (GRU) model is the best model on average and improves the average $CC_{norm}$ values obtained by the previous state-of-the-art model (2D-CNN) by about 10%, a considerable amount given the difficulty of the task and the noisiness of auditory neurons.

StateNet models (but also the other recurrent alternative, DNet) do not need to process auditory inputs over a time window of fixed duration but instead gradually build their predictions of neural activity at the current timestep from past stimulus inputs (see Fig. 3). Thereby, they implicitly learn the extent of their temporal integration window without the need for any extra hyperparameter. It makes them biologically more plausible as durations used in stateless models usually exceed the upper boundary of biological axonal delay distribution (around 50 ms, see Rahman et al.[25]). Interestingly, they also outperform Transformers at fitting auditory neural responses, even if this architecture is now used as a gold-standard in many complex machine learning tasks[47,55–57].

If our work is the first to use recurrent deep neural networks to characterize single unit responses in auditory cortex, there were several studies that introduced some form of recurrence into stateless models to capture adaptation mechanisms. To cite a few examples, David and Shamma[7] proposed a nonlinear stimulus-response model that accounted for synaptic depression.[26] used a nonlinear pre-processing stage based on a high-pass filtering to replicate adaptation to mean-sound level in the auditory midbrain. In Rançon et al.[27], a pair of parameterized temporal filters permitted to capture adaptive mechanisms within ON and OFF responses. All these models led to substantial improvements (although not as pronounced as in the present study) in terms of neural responses fitting performances, thereby reinforcing the idea that recurrent mechanisms are important to characterize auditory responses. Recurrence in these approaches was nonetheless obtained from various add-ons (i.e., nonlinearities) to the original STRF-based models and thus did not constitute a core architectural principle, as in the present study. Temporal recurrence appears to serve as a more effective computational mechanism across both short and long time scales, in agreement with findings from previous neurophysiological studies[23,53,54]. Indeed, our experiments with TBPTT demonstrate that our StateNet models are capable of predicting current neural activations based on auditory information from the distant past. The earliest time steps appear to make a contribution that, while moderate, is significant enough to reach state-of-the-art performances on the tested datasets.

One possible reason for the better $CC_{norm}$ obtained with our gated StateNet models is that they can reproduce many forms of adaptive mechanisms described in the literature, while stateless approaches (including Transformers) cannot. We provide in the Supplementary materials (see "Mathematical mapping of RNN networks to adaptation mechanisms observed in the auditory system") a mathematical demonstration that gated StateNet models can approximate SSA[5], STP[7,8–10] CGC or context dependence[11,12]. If these computational mechanisms operate at different time scales[7], they might however be complementary[29] and the protean nature of StateNet models offers a unique framework to implement this variety of functions all at once. Among our StateNet models, the GRU probably offers the best compromise between task accuracy, conceptual simplicity (it only requires 4 equations instead of 5 for the LSTM and has a lighter formalism than S4 and Mamba) and implementation efficiency (it is natively supported in PyTorch). However, we also find SSMs particularly attractive as the abundance of mathematical tools accompanying them can ease the reverse-engineering process. Now that we have empirically demonstrated the efficiency of StateNet models, we hope that future research in computational neuroscience will leverage their capabilities.

Recurrent networks are currently the subject of intensive research in the AI community[58,59], and we anticipate that in the near future, new models should permit to even fit neural responses better and thereby to improve our understanding of the associated neural mechanisms.

Because StateNet models do not have explicit spectro-temporal weights, classical methods to estimate the auditory selectivity of their neural units are not applicable[22,30]. To circumvent this issue, we proposed here an approach based on advances from the deep learning community[38–40] and which leverages automatic differentiation in recurrent networks. This approach permits for each neural unit to extract its nonlinear receptive fields and to estimate the auditory features that maximize its responses (see Fig. 3). It generalizes linear STRFs for any (including nonlinear and recurrent) models, is easy to compute, and does not introduce any additional hyperparameter. Like STRFs derived from the STA, GradMaps are dependent on their model's parameters and, consequently, on the stimulus set used to fit them. (see Woolley et al.[60]). However, natural stimuli have been shown to yield higher explanatory power than artificial stimuli (such as white noise, TORCs, etc.) in terms of better generalization to other stimulus classes[18,61] so it is logical to focus on the latter.

One important property of our approach is that it allows for recurrent models to reveal neural selectivity far beyond the temporal windows classically used to estimate STRFs, in line with previous electrophysiological studies, which showed that some auditory neurons integrate their inputs over timescales of a few seconds[5,11]. As an illustration, Supplementary Fig. S3 shows that for many StateNet models, sounds presented more than 2 s before the neuron's response still modulate the response strength. Thus, with these observations, our models and tools generate new hypotheses to be tested experimentally.

Our proposed approach is not the first attempt to directly model the nonlinear stimulus-response mappings of auditory cortex neurons with deep neural networks. In a recent work, Keshishian et al.[30] defined STRFs as the derivative of the output with respect to the input (i.e., the data Jacobian matrix, similar to our GradMaps, see Wang et al.[38]) in this context. Their method allowed them to find the mathematically equivalent linear function that the neural network applies to each instance of the stimulus, thereby producing linear STRF windows dynamically changing at each time step (DSTRFs) and still restricted to a fixed duration. Moreover, this method assumes strong constraints on the neural network (a feedforward architecture with ReLU activations, no biases in the intermediate layers, as well as a linear output) and necessitates simplifying it into an equivalent multilayer perceptron. Our approach does not rely on any temporal window, is totally agnostic on the network architecture (in the present study, it is notably applied on StateNet models but also on more classical models and on Transformers), and thus constitutes a much broader framework to interpret the nonlinear receptive fields learned by a model.

We emphasize that our Dreams/GradMaps procedure is explicitly optimized to identify stimuli that maximize a unit's predicted response. While this approach effectively reveals excitatory features, it may overlook modulatory or suppressive effects that depend on the broader stimulus context. Contextual suppression and long-lasting context dependence have been reported in auditory cortex[5,11], and these phenomena could reduce a neuron's response without altering the optimal excitatory pattern discovered by gradient-based maximization. To address this issue, a straightforward extension of our work would be to compute GradMaps after different preceding contexts (for example, conspecific vocalizations, silence, or noise) and compare them directly to baseline (no-context) GradMaps. Systematic differences would reveal how context shifts the neuron's input-output function (suppression, gain modulation, or changes in latency). One can also imagine inverting the optimization objective (optimize for minimal responses or for maximal difference between two contextual conditions) to explicitly target context-dependent suppression. Such model-driven contrasts can be validated experimentally in closed-loop paradigms. These analyses would complement the present results and sharpen hypotheses about context-dependent suppression and facilitation in auditory neurons. Our tool can also be easily extended to the population level by optimizing the

responses of a group of neurons instead of a single unit, in order to better understand interactions between sensory cortical units.

Over the last decades, electrophysiologists have estimated the preferred stimulus of a given neural unit using white noise[17], synthetic stimuli[62], or natural sounds[18]. As the performances of neural encoding models and methods to estimate sensory selectivity progress, it now becomes possible to optimize the features of the stimulus directly during the experiments. Such close-loop approaches have recently been conducted on the visual modality with order-zero optimization methods such as genetic algorithms[63,64] but also gradient-based methods[65,66]. Because it is easy to compute and can be applied to any (including recurrent) architecture, the method proposed in the present study could permit to improve these close-loop approaches and to extend them to dynamic stimuli like sounds but also videos. Thus, our work constitutes an important step toward a better characterization and control of neural populations at the unit level, and reaffirms the interest of a hybridization between computational neuroscience and AI on this avenue.

## Methods

We begin this section with a summary of common models of auditory neural responses and introduce a novel, Transformer-based architecture, as well as our fully recurrent model called StateNet. Then, we describe the electrophysiology datasets used in this study and set the mathematical framework of the neural response fitting task. Because conventional models and most datasets were already described in a prior study from our group[67], we invite readers to consult it for a more precise description of conventional models and datasets. Lastly, we explain our process for reverse-engineering models, which generalizes STRFs for arbitrary network depth, width, and degree of nonlinearity.

### Canonical computational models of auditory neural responses

The aim of computational models of neural responses in auditory cortex is to convert ("encode") incoming sound stimuli into time-varying firing rates/probabilities that predict electrophysiological measurements made in auditory areas. Traditionally, these models use the cochleagram of the stimulus—a spectrogram-like representation that mimics processing in the cochlea—and are rate-based (as opposed to spiking). Many such models have been proposed in the literature, ranging from simple Linear ("STRF" or "L") approaches[17,18] to more complex methods based on multi-layer CNNs[31]. However, all these models use temporal convolutions with finite window lengths and, therefore, finite TRFs. In this case, the duration of the TRF is a hyperparameter that is arbitrarily defined by the modeling scientist.

More formally, a model $\mathcal{M}$ is a causal application $\mathbb{R}^{F \times T} \mapsto \mathbb{R}^{N \times T}$ where $F$ is the number of frequency bands of a stimulus spectrogram, $T$ a variable number of time steps, and $N$ a number of units/channels whose activity to predict.

In this paper, we use five models based on this approach: the Linear (L) model[17,18], the Linear-Nonlinear (LN) model[45,68], the Network Receptive Field (NRF) model[22], the Dynamic Network (DNet) model[25], and finally a deep 2D-CNN model[31]. A general architecture of a convolutional model is illustrated in Fig. 7a. Because a full description of these approaches was already provided in previous benchmarks[27], we only review here some of their shared general features. We also report minor modifications that we introduced in their implementations so as to make them work on our unified PyTorch pipeline. We invite the reader to consult the original studies that introduced these models for a more detailed description of their functioning.

### Auditory periphery

The initial processing stage converts the stimulus sound waveform into a biologically plausible spectrogram-like representation $x \in \mathbb{R}^{F \times T}$, thereby reflecting operations realized by the cochlea. In the literature, the waveform-to-spectrogram transformation can be performed through a simple short-term Fourier decomposition, or more often through temporal convolutions with a bank of mel or gammatone filters that are scaled logarithmically along the frequency axis. Following the latter, a compressive function such as a cubic root or

logarithm is applied. Although the combination of both of these operations makes a consensus, there is a variability across studies in their implementation. However, it was shown that such variations are all more or less equivalent and still provide good cochlear sound encodings when modeling higher-order auditory neural responses. As a result, simple transformations should be preferred[69]. In order to facilitate present and future comparisons with previous methods, and to limit as much as possible the introduction of biases due to different data pre-processings, we directly use here the cochleagrams provided in each dataset.

### Core principle

Classical models rely on a cascade of temporal convolutions with a stride of 1 performed on the cochleagram of the sound stimulus, interleaved with standard nonlinear activation functions (e.g., Sigmoid, LeakyReLU) and followed by a parametric output nonlinearity with learnable parameters (e.g., baseline activity, slope saturation value). In all models, the cochleagram is systematically padded to the left (i.e., in the past) with zeroes prior to the temporal convolution operations, in order to respect causality and to output a time series of neural activity with as many time bins as in the input cochleagram.

### Single unit vs. population fitting

In datasets where all sensory neurons were probed using the same set of stimuli, it is possible for computational models to predict the (vector) activity of the whole population[31]. This population coding paradigm allows to train a single model with some learnable parameters shared across all neural units under consideration, and some specific to each unit. As a result, the common backbone tends to learn robust and meaningful embeddings, which further reduces overfitting. Performances are, on average, better across the population than when fitting an entire model for each unit. Furthermore, this process drastically reduces training time and brings it down to a computational complexity of $O(1)$ instead of $O(N)$, where $N$ is the total number of units in each dataset. For these reasons, we adopt the population coding paradigm whenever possible, that is, when various single-unit responses were recorded for the same stimuli. This is the case for the NS1, NAT4-A1, NAT4-PEG, AA1-MLd and AA1-Field_L datasets.

### Output nonlinearity

All but the L model were equipped with a parametric nonlinear output activation function, learned alongside all other parameters through gradient descent. We used the following 4-parameter double exponential:

$$f(x) = a \exp(-\exp(kx - s)) + b \qquad (1)$$

where $b$ represents the baseline spike rate, $a$ the saturated firing rate, $s$ the firing threshold, and $k$ the gain[21,31]. Importantly, in the case of population models (i.e., when predicting the simultaneous activity of several units, see below), each output neuron learns a different set of these four parameters.

### Regularization and parameterization

Canonical models based on convolutions are prone to overfitting, and many strategies were proposed to limit this effect, such as the parameterization of spectro-temporal convolutional kernels[21,60,68,70]. To stick to the most extensively reviewed version of these canonical models, as well as to highlight their limitations, our implementation did not include any such methods. Furthermore, we did not use any data augmentation techniques, weight decay, or dropout during training, as it was previously shown that such approaches complexify training and yield little to no improvements in performances[27,31]. Instead, we used Batch Normalization (BN), which greatly improved the robustness and performances of all models, including the linear one (L), without compromising its nature, as after training, BN's scale and bias terms can be absorbed by the models own weights and biases.

### Transformer model

Over the last years, attention-based Transformer architectures[47] have been more and more used by the AI community as an alternative to RNN for

modeling long sequences[57], from text[56] to images[55]. Contrary to stateful approaches, which rely on BPTT for training, Transformers do not suffer from vanishing or exploding gradients. However, they present some drawbacks, such as quadratic algorithmic complexity scaling with the sequence length. To investigate whether this model is well-suited for fitting dynamic neural responses in the auditory cortex, we developed a novel architecture based on the attention mechanism (see Fig. 7b). To the best of our knowledge, it is the first model of its kind that is proposed for this task.

As for stateless models, a hyperparameter $T$ defines the length of the temporal context window that serves to predict single-unit or population activity at the current time step. Within this window, the spectrogram of the auditory stimulus is projected into $T$ tokens (one per time step) of embedding size $E$ by means of a fully connected layer applied to each frequency vector. A learnable positional embedding is subsequently added to this compressed spectrogram representation before feeding it to a Transformer encoder with 1 layer, 4 heads, and a dimensionality of 48[47]. These hyperparameters were fixed for all datasets. As the outputs of the Transformer encoder are given as $T$ processed tokens, we apply global average pooling over the token dimension and use the resulting tensor as the input to a final fully-connected readout layer, followed by a double-exponential activation function with per-unit learnable parameters. We observed empirically that the global average pooling operation is crucial to reach good performances while drastically reducing the size of the last fully connected layer.

### StateNet models

A high-level schematic of the processing realized by our StateNet models is provided in Fig. 7c. Their architecture can be decomposed into three main elements: a downsampling layer, a stateful bottleneck, and a readout.

**Downsampling locally connected (LC) layer.** At each time step, we downsample the current vector of spectral information to reduce the dimensionality of input stimuli. Contrary to natural images, which are shift-invariant, spectrograms have very different statistics between low and high frequencies, thereby making weight sharing a less efficient computational strategy. Further motivated by the tonotopic organization observed along the auditory pathway[71,72], we use a LC layer with restricted receptive fields (as for convolutional layers) but with independent weights across frequency bands (see Supplementary Fig. S2). In other words, LC contains a subset of the weights of a fully connected (FC) layer, defined by a convolutional (CONV) connectivity pattern over the frequency axis. In theory, the performances obtained with this LC scheme are only a lower bound of what can be reached with FC. However, we found in practice that LC yields overall only slightly lower or similar results using the same hyperparameters (see Supplementary section "Ablation study: connectivity in the first layer of StateNet"), but with a smaller number of free learnable parameters, hence reducing the risks of overfitting and permitting a better generalization. LC also outperformed the CONV approach because it relaxes the weight-sharing constraint.

Despite its biological and computational motivations, this approach has only rarely been incorporated into models of auditory processing. For example, Chen et al.[73] also used a local connectivity for speech recognition, but weight kernels were 2d (spectro-temporal) instead of 1d (only spectral and shared across the temporal dimension). In the field of computational neuroscience, Khatami and Escabí[74] imposed local Gaussian kernels as fully connected weights. Our implementation differs in that it implements the trade-off between CONV and FC, with fewer parameters than FC, and possibly faster execution.

All in all, our proposed LC downsampling scheme is more biologically plausible than the FC and CONV alternatives, while providing a better trade-off between performances at the neural response fitting task and model complexity. In addition, it executes faster than the FC approach and prior LC implementations. An optimized PyTorch module is available on our code repository.

**Stateful bottleneck(s).** It is composed of a single layer of either type of RNN, as adding more layers did not seem beneficial to performances in our preliminary experiments. RNNs are a type of artificial neural networks specifically developed to learn and process sequential inputs and notably temporal sequences. They work iteratively and build their output at each timestep from the current inputs as well as a constantly updated internal representation called hidden state. Because the mathematical details of the modules used here are fully provided in previous studies, we only report below their main properties. We invite the readers to consult the associated papers if a more thorough understanding of their computational principles is needed.

**Vanilla RNN.** In this paper, we designate "vanilla RNN" as the classical Elman network[48] natively implemented in PyTorch, often considered as the most naive implementation of this class of models.

**Gated RNNs: LSTM and GRU.** A notorious problem with vanilla RNNs occurs when dealing with long sequences, as gradients can explode or vanish in the unrolled-over-time network[75,76], preventing them from exploiting long-range dependencies and therefore from performing well on large time scales. Gated RNNs such as LSTM[49] and GRU[50] successfully circumvent these difficulties, and have imposed themselves as efficient modules for learning sequences in a recurrent approach.

**State-space models (SSMs): S4 and Mamba.** State Space Models (SSM) is a new class of models that takes inspiration from other fields of applied mathematics, such as signal processing or control theory[77]. Specifically designed for sequence-to-sequence modeling tasks (and thus for time series prediction as in the present study), they build upon the State-Space equations below with various parameterization techniques and numerical optimization methods.

$$\begin{cases} \dot{x}(t) = Ax(t) + Bu(t) \\ y(t) = Cx(t) + Du(t) \end{cases} \tag{2}$$

where $u \in \mathbb{R}$ is the input, $x \in \mathbb{R}^N$ the hidden state vector, $y \in \mathbb{R}$ the output, and $A$, $B$, $C$ and $D$ system matrices.

In particular, the original *Structured State-Space Sequence (S4)* model appears as one of the simplest versions of this paradigm, with only a few constraints on the system[51]. At the opposite, the *Mamba* architecture is one of the more recent and sophisticated propositions in which system matrices are input-dependent, and has been shown to perform on par with Transformers on various benchmarks[52]. As a whole, SSMs hold the promise of data-scalable models with great performances, while benefiting from a solid theoretical foundation, which permits to connect them to convolutional models (CNNs), RNNs with discrete timesteps, but also continuous linear time-invariant systems of ordinary differential equations. This last property is particularly interesting as it can ease the reverse-engineering process of a fitted model, thereby allowing for high levels of interpretability. In addition, trained SSMs can easily be modified to work at any temporal resolution, opening interesting use cases for the field of computational neuroscience and neural engineering.

**Readout.** The readout neural activity for a given unit is computed from a linear projection of the output state vector into a single scalar, repeated at each timestep.

### Electrophysiology datasets of auditory responses

To characterize the ability of models to capture responses in the auditory cortex, we fitted them in a supervised manner on a wide gamut of natural audio stimulus-response datasets. These datasets were collected in different species (ferret, rat, zebra finch) and brain areas (MGB, AAF, A1, PEG, MLd, Field L), under varying behavioral conditions (awake, anesthetized) and using different recording modalities (spikes, membrane potentials). All of them are freely accessible on online repositories and were used with respect

to their original license. Because the *"NS1"*[22], *"NAT4"*[31,78], and *"Wehr"*[24,44] datasets have already been used in a previous study from our group and their pre-processing pipelines have been extensively described in the associated article[27], we only describe here the new datasets added to the current study. The corresponding data pre-processing methods are representative of what was performed in the previous datasets.

**AA1 datasets: MLd, field L (zebra finch).** These two datasets consist in single-unit responses recorded from two auditory areas (MLd and Field L) in anesthetized male zebra finches, by Frederic Theunissen's group at UC Berkeley[43,46,79]. Stimuli were composed of short clips (<5 s) of con-specific songs to which animals had no prior exposure, and were modeled by log-compressed mel spectrograms with 32 frequencies ranging 0–16 kHz, at a temporal resolution of 1 ms. Extracellular recordings yielded a total of 50 single units in each area after spike sorting. Spike trains were binned in non-overlapping windows of 1 or 5 ms, matching the resolution of the stimulus spectrogram. Sounds were presented with an average of 10 trials, and the PSTHs were obtained for each neuron and stimulus after averaging spike trains across trials and smoothing with a 21 ms Hanning window[26,80]. For each neuron, recordings were performed in response to the same 20 audio stimuli, thereby allowing the training of a single model to predict the simultaneous activity of the whole population (i.e., a *"population coding"* paradigm).

These data, also referred to as *"AA1"*, are freely accessible from the CRCNS website (https://crcns.org/data-sets/aa/aa-1/about) and were used with respect to their original license.

**Asari dataset: A1, MGB (rat).** This dataset consists in single-unit responses recorded from primary auditory cortex (A1) and medial geniculate body (MGB) neurons in anesthetized rats, by Anthony Zador's group at Cold Spring Harbor Laboratory[11,44]. Stimuli were natural sounds typically lasting around 2–7 s, and originally sampled at 44.1 kHz and then resampled at 97 kHz for presentation. Their associated cochleagrams were obtained using a short-term Fourier transform with 54 logarithmically distributed spectral bands from 0.1 to 45 kHz, whose outputs were subsequently passed through a logarithmic compressive activation. The temporal resolution of stimulus cochleagrams and responses was set to 5 ms. Because for each cell, recordings were performed in response to a different set of probe sounds, we could not apply the population coding paradigm, and one full model was fitted on the stimulus-response pairs for each unit. Contrary to the other datasets, recordings here are intracellular membrane potentials obtained through whole-cell patch-clamp techniques. One remarkable feature of these data is their very high trial-to-trial response reliability, making the noise-corrected normalized correlation coefficients of model predictions almost equal to the raw correlation coefficients (see "Performance metrics" subsection).

Despite the good signal-to-noise ratio of this dataset, some trials are subject to recording artifacts and notably to drifts, which may be caused by motion of the animal and/or of the recording electrode, or electromagnetic interferences with nearby devices. Note that drifts do not contaminate supra-threshold signals resulting from spike-sorted activity, such as PSTHs, because they are strictly positive and thus have a guaranteed stationarity. In order to remove these drifts, we detrended all responses using a custom approach further explicited in Supplementary section "Detrending with MedGauss filter".

Similar to the previous dataset, these data can be found on CRCNS website (https://crcns.org/data-sets/ac/ac-1) as a subset of the *"AC1"* dataset.

## Neural response fitting task

Because the current benchmark directly builds upon a previous study conducted by our group[27], performances and model trainings were conducted according to the same methods, which we describe again below.

**Task definition.** Neural response fitting is a sequence-to-sequence, time series regression task, taking a spectrogram representation $x \in \mathbb{R}^{F \times T}$ of a sound stimulus as an input, and outputting several 1d time series of neural response (one for each unit), $\hat{r} \in \mathbb{R}^{N \times T}$. As the latter, we use the Peri-Stimulus Time Histogram (PSTH), which is the average recorded neural response across repeats. The loss function is the mean squared error (MSE) between the predicted time series and the recorded PSTH, and was evaluated for each time bin of each sequence:

$$\mathcal{L} = \frac{1}{NT}\sum_{n=1}^{N}\sum_{t=1}^{T}(\hat{r}_n[t] - r_n[t]) \quad (3)$$

where $\hat{r}_n[t]$ is the predicted neural response for neuron $n$ at time-step $t$, $r_n[t]$ the corresponding PSTH, $N$ is the total number of recorded neurons to fit, and $T$ is the total number of time-steps in the time series (to simplify the notations, we drop the time dependencies symbols $[t]$ thereafter).

**Performance metrics.** The neural response fitting accuracy of the different models is estimated using the raw correlation coefficient (Pearsons' $r$), noted $CC_{raw}$, between the model's predicted activity $\hat{r}$ and the ground-truth PSTH $r$, which is the response averaged over all $M$ trials $r^{(m)}$:

$$r = \frac{1}{M}\sum_{m=1}^{M}r^{(m)} \quad (4)$$

$$CC_{raw} = \frac{Cov(r, \hat{r})}{\sqrt{Var(r)Var(\hat{r})}} \quad (5)$$

where the covariance and variance are computed along the temporal dimension. Assuming neural variability is purely noise and given a limited number of stimulus presentations, perfect fits (i.e., $CC_{raw} = 1$) are impossible to get in practice. In order to give an estimation of the best reachable performance given neuronal and experimental trial-to-trial variability, we use here the normalized correlation coefficient $CC_{norm}$, as defined in refs. 80[,81]. For a given optimization set (e.g., *train*, *validation* or *test*) composed of multiple clips of stimulus-response pairs, we first create a long sequence by temporally concatenating all clips together. We then evaluate the signal power $SP$ in the recorded responses as:

$$SP = \frac{Var(\sum_{m=1}^{M}r^{(m)}) - \sum_{m=1}^{M}Var(r^{(m)})}{M(M-1)} \quad (6)$$

which allows to compute the normalized correlation coefficient:

$$CC_{norm} = \frac{Cov(r, \hat{r})}{\sqrt{SP \times Var(\hat{r})}} \quad (7)$$

When only one trial is available, we set $CC_{raw} = CC_{norm}$, which corresponds to a fully repeatable recording uncontaminated by noise, thereby preventing any overestimation of performances by setting a lower bound in the absence of data.

**Optimization process/model training.** All models were randomly initialized using default PyTorch methods and trained using gradient descent and backpropagation. Time recurrent models (DNet and State-Nets) were trained using BPTT. Each training sample was a full stimulus-response pair whose duration varied between datasets because of the different recording protocols, but also within some datasets (AA1, Wehr, Asari) in order to maximize the amount of trial information for evaluation. We used AdamW optimizer[82] and its default PyTorch hyper-parameters ($\beta_1 = 0.9$, $\beta_2 = 0.999$). We used a batch size of 1 for all datasets except NAT4, which have a limited number of training examples, and a batch size of 16 for both NAT4 datasets, which have consequently more (see *"Supplementary Note 6: Dataset and model details"*). The learning

rate was held constant during training and set to a value of $10^{-3}$. We found empirically that these values led to better results.

We split each dataset into a training, a validation, and a test subset, respecting a 70–10–20% ratio as much as possible, depending on the number of stimulus-response pairs available for each cell in each dataset. After each training epoch, models were evaluated on the validation set, and if the validation loss had decreased with respect to the previous best model, the new model was saved. Models were trained until there was no improvement during 50 consecutive epochs on the validation set, at which point learning was stopped, the last best-performing model was saved, and evaluated on the test set. This procedure was repeated 10 times, each corresponding to a random seed, for different train-valid-test data splits and model parameters initializations, and the test metrics were averaged across splits. All models going through the exact same training pipeline (i.e., waveform-to spectro-gram transform, training hyperparameters, etc.) ensured fair comparison between them, implying that architectures with higher test accuracy are genuinely better, despite potential differences with their original studies.

### Truncated backpropagation through time

With regular BPTT, the entire RNN model is unrolled back in time, creating a graph that grows in size linearly with the sequence length. In addition, for most tasks, old time steps are less informative of the present than the most recent ones, and gradients can either vanish or explode (i.e., "vanishing/ exploding gradient problem")[75]. TBPTT is a variation of this training algorithm aiming to alleviate these computational constraints[83]. Its principle relies on removing from the graph time steps older than a fixed temporal horizon $K_2$. If the loss is evaluated every other $K_1$ time steps, we note this algorithm $TBPTT(K_1, K_2)$. Fig. 8. illustrates the underlying computations. In this study, in order to maximize the number of samples used for training, the loss is evaluated and backpropagated every time step, and therefore $K_1 = 1$. As a result, we simplify notations by referring to $K_2$ as $K$. For the first $t < K_2$ time steps, the graph is built from the start of the sequence. For generality, the regular BPTT algorithm used to train our models in the main experiments corresponds to $TBPTT(K_1 = 1, K_2 = T)$, $T$ being the sequence length. We also distinguish two sub-cases of TBPTT:

- *TBPTT with warmup*. The model is initialized with the default (null) hidden state $h_0 = 0$ at the very start of the sequence (see Fig. 8a). Inputs are then processed sequentially without building the computational graph, up to the $K$ last time steps before loss evaluation. We qualify these first steps as a "warmup". As a result, the graph starts with a model in an intermediary, non-default, and ecological hidden state resulting from this procedure.
- *TBPTT without warmup*. Here, warmup steps are skipped and the model is directly initialized to the default hidden state at the $K$th time step before loss evaluation (see Fig. 8b). Therefore, the model here does not have access to *any* prior information at all, making it a fairer comparison to the training of stateless models.

### Model interpretability with feature visualization: gradient maps and deep dreams

We propose here a gradient-based iterative method which, for each unit of a trained neural network $\mathcal{M}$, extracts its nonlinear receptive fields and estimates the auditory features $x_i$ that maximize its responses. This method builds on feature visualization techniques originally introduced in the AI community[38–40] and is known as gradient ascent, leveraging the fact that all the mathematical operations in our models are differentiable. The different steps of this approach are illustrated on Fig. 3a and can be summarized as follows:

1. As the first input to the model, use the null stimulus $x_0 \in \mathbb{R}^{F \times T}$, a uniform spectrogram of constant value (0 in our case). This initial stimulus is unbiased and bears no spectro-temporal information. From an information-theoretic perspective, it has no entropy. For a parallel with electrophysiology experiments, it is worth noting that the spectrogram of white noise is theoretically uniform too. The absence of spectro-temporal correlations within probe stimuli (which we respect

here) is a strong theoretical requirement that led to the use of white noise in the initial development of the linear STRF theory[17], while further studies preferring natural stimuli used advanced techniques to correct their structure[18].

2. Pass $x_0$ through the model and compute its outputs (i.e., the predicted time-series of activation for the whole neural popula-tion): $\hat{r} = \mathcal{M}(x_0) \in \mathbb{R}^{N \times T}$.
3. Define a loss $\mathcal{L} : \mathbb{R}^{N \times T} \mapsto \mathbb{R}$ to minimize and compute it from the model prediction $\hat{r}$. In this paper, we only targeted a single unit $n$ and used the opposite of its activation at the last ($T$th) time-step: $\mathcal{L}(\hat{r}) = -\hat{r}_n[T]$. To compute the STRF associated with any neural population $\mathcal{N}$, the following general loss can be used: $\mathcal{L}(\hat{r}) = -\frac{1}{Card(N)} \sum_{n \in \mathcal{N}} \hat{r}_n[T]$. The choice of this loss function permits to make the connection with the Spike-Triggered Average (STA) approach used by electrophysiologists, and where the stimulus instances preceding the discharge of the target unit are averaged[17,41,42]. Models in our study were fitted to PSTHs or membrane potentials and thus output floating point values, which can be viewed as a spike probability. Trying to maximize this value at the present time-step (the last of the time series) by constructing previous stimulus time steps closely mimics STA. Maximizing the average firing rate across the whole stimulus presentation could be interesting to investigate in future studies.
4. Back-propagate through the network the gradients of this loss $g_0 = \frac{\partial \mathcal{L}}{\partial x_0}$, thereafter referred to as GradMaps. From their definition, these GradMaps can be directly related to linear STRF (see Supplementary text "Bridging the gap between STRFs, gradMaps and dreams: theoretical framework").
5. Use these gradients to perform a gradient ascent step and modify the input stimulus. For simplification, we denote this operation as $x_1 = x_0 - \alpha g_0$, but some optimizers have momentum and more ela-borate update rules. This is notably the case of Adam and AdamW[82], which were used in our study. We did not use the SGD optimizer as it converged to much higher loss values—so less optimal—in preliminary experiments.
6. Repeat these steps until an early stopping criterion is satisfied, in our case after a fixed number of iterations (1500, a value which led to sufficient loss decreases, see curves in Fig. 3). The result of this process is an optimized input spectrogram $d = x_i$ that maximizes the activation of the target unit(s), thereafter referred to as Dream. If we define $\mathcal{F} : \mathbb{R}^{F \times T} \mapsto \mathbb{R}^{F \times T}$ as one iteration of the above process, such that $x_i = \mathcal{F}(x_{i-1}|\mathcal{M}, \mathcal{L}, n)$ then recursively get $x_i = \mathcal{F} \circ \mathcal{F} \circ \cdots \circ \mathcal{F}(x_0|\mathcal{M}, \mathcal{L}, n) = \mathcal{F}^i(x_0|\mathcal{M}, \mathcal{L}, n)$.

This approach does not assume any requirement on the model to interpret. It can produce infinitely long GradMaps, which are relatable to linear STRFs and can be implemented on any architecture, including RNNs like StateNet, but also stateless and transformers.

### Dream and GradMap energy

A trace of temporal integration for a model can be simply defined from its GradMap $g$ as the mean over all frequency bands $f$ of squared elements for each latency $t$. We designate this measure the "Energy" of the GradMap:

$$E[t] = \frac{1}{F} \sum_{f=1}^{F} g[f, t]^2 \tag{8}$$

### GradMap similarity matrix

As shown in the corresponding results section, this matrix aims to identify functional clusters of models based on their GradMaps; it is built using the following methodology.

The GradMaps of the models to compare are first computed with a number of time steps $T$ slightly above their theoretical TRF size. In the case

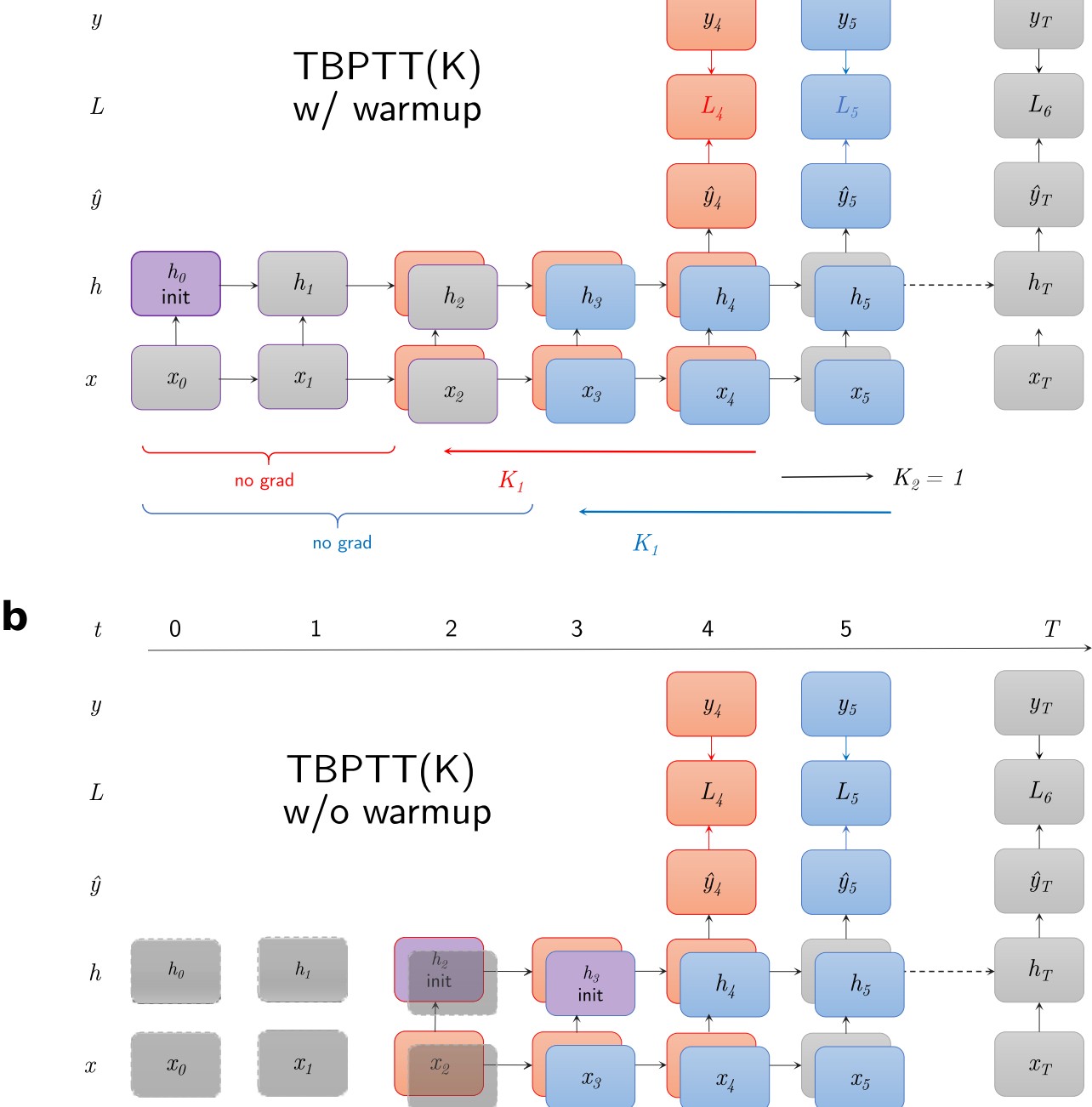

**Fig. 8 | Truncated backpropagation through time (TBPTT): methods. a** With warmup, the initial hidden state (purple) is the first of the training sequence. Subsequent time steps (in gray, delimited by "no grad") correspond to the warmup during which the hidden state is updated. In this example, loss evaluation is shown for two time steps (4 and 5), and their respective graph are colored in red and blue. $K_1$ can be interpreted as the temporal stride between two loss evaluations and $K_2$ (here, 3) as the maximum graph length, in number of time steps. **b** Without warmup, the first time steps are skipped and do not belong to the computational graphs. Therefore, the latter starts from the default, null hidden state instead of an intermediary value resulting from the warmup procedure above.

of the present study, stateless models had a TRF size of up to 43 time steps. Therefore, we computed GradMaps of $T = 50$ time steps for all models, including StateNet. The length of the GradMap should not be much greater than the theoretical TRF size of stateless models; otherwise, correlations between the GradMaps of the latter would artificially tend towards 1, as time steps beyond the receptive field do not receive any gradient and remain unaffected and at their initial value of 0. This step yields one GradMap per neuron, dataset, and model. After a flattening operation into a $F \times T$ vector, we compute Pearson's correlation coefficient as a pixel-wise metric to compare how similar GradMaps are between models, for a given neuron and dataset. The choice of the CC here instead of other distance metrics (e.g., Euclidean) is motivated by the fact that we are comparing the overall structure of the GradMap/STRF (e.g., how inhibitory and excitatory regions are placed relative to each other) rather than its value. Furthermore, the goodness-of-fit of the Linear STRF model, to which we relate the GradMap, is insensitive to changes in scaling and shifting, precisely because the primary evaluation metric in the neural response modeling community is based on the CC too. As a result, if two models present the same GradMap

**Article**

up to an affine transformation, their functional similarity should be classified as perfect, which would not be the case with metrics, such as a pixelwise MSE.

## Statistics and reproducibility

One major contribution of this work is the sheer amount of implemented models and compiled datasets. All models were implemented and trained using PyTorch, a gold-standard library for deep learning in Python, leveraging autodiff. Datasets were pre-processed under the same format of a PyTorch Dataset class for convenience. Jobs required less than 2 GiB and were executed on Nvidia Titan V GPUs, taking tens of minutes to several hours, depending on the complexity of the model. As an example, the population training (5 seeds) of the StateNet GRU model trained in population coding on all 73 NS1 neurons in parallel typically takes less than 10 min. Conversely, the single unit training (5 seeds) of the same model on the 21 neurons of the Wehr dataset takes more than 3 h.

## Reporting summary

Further information on research design is available in the Nature Portfolio Reporting Summary linked to this article.

## Data availability

All datasets used in this study are publicly available online and were used with respect to their original license. They can respectively be found on the CRCNS website (https://crcns.org), Zenodo (https://zenodo.org/records/7796574), and the Open Science Framework (OSF) website (https://osf.io/ayw2p/).

## Code availability

Instructions to download and pre-process datasets, the source code of our models, as well as training and evaluation scripts are freely available on our GitHub repository: https://github.com/urancon/deepSTRF. It will complement the already existing basis of this repository, originally developed in a previous work of our group[27]. The aims of this library are to normalize practices in the field of neural response fitting with quantitative computational models, to provide a common ground for benchmarking and to permit a fair comparison between models on various preprocessed datasets, with ready-to-use and user-friendly classes and methods.

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

## Acknowledgements
This study was supported by a grant from the Agence Nationale de la Recherche (ANR-21-CE28-0021, ANR PRC ReViS-MD) and by a FLAG-ERA funding (Joint Transnational Call 2019, project DOMINO), both awarded to BRC.

## Author contributions
U.R. developed and compiled models and datasets. U.R., T.M. and B.R.C. conceptualized the experiments. U.R. coded and conducted them. U.R., T.M. and B.R.C. analyzed and interpreted the results. U.R. created the Figures with feedback from T.M. and B.R.C. All three authors participated in the writing of the manuscript.

## Competing interests
The authors declare no competing interests.
