## [Transparent Peer Review file · Communications Biology]

Temporal recurrence as a general mechanism to explain neural responses in the auditory system

Corresponding Author: Mr Ulysse Rancon

Version 0:

Reviewer comments:

Reviewer #1

(Remarks to the Author)

The paper introduces a new class of computational models known as "StateNet," which are based on deep recurrent neural networks (RNNs). These models demonstrate better performance than previous approaches in fitting neural responses within the auditory system. One key advantage of StateNet models is that they do not require predefined time windows to process auditory stimuli, unlike earlier stateless methods and transformer models. The authors' approach indicates that auditory neurons integrate information over much longer timescales—several seconds—than what current spectro-temporal receptive field (STRF) techniques suggest. I find the paper interesting; however, there are some details and comments that need to be addressed before publishing.

Regarding the introduction:

1) Could you give a definition, and a corresponding citation, of what you call xAI? It is a new term whose definition can be a bit vague.

Regarding the methods:

1) The text does not provide detailed information about the size (e.g., number of layers, neurons per layer) or specific characteristics of each neural network model (e.g., GRU, LSTM, Transformer). The work needs a table that compares which ones and how many parameters each of the neural networks (models) has. From reading the work, I do not understand how many input units the models have, nor how many units the hidden layer has according to the case. I understand that the output layer will have as many units as the corresponding data set. I would also like to know if the models that have fewer units at the output, corresponding to each data set, present a better accuracy or not, (to understand what happens with the dimensionality of the system).

2) I would like you to better explain aspects of the training such as the temporal length of the spectrogram segments you use.

3) What happens if we analyze the dynamics from the point of view of the connectivity matrix and the activity produced by the artificial units, prior to the output layer, (the case of the vanilla RNN network) Can any structure be observed in the PCA component space (such as, for example, trajectories observed in the works that study motor control)? Can this work be compared with any of those works?

Results:

The first time the dataset appears, it does not have the corresponding citation to be able to easily locate the origin of the database.

It is not very clear to me how the results are shown in Figure 1. The concept of PSTH (Peri-Stimulus Time Histogram) is not explicitly defined in the text, but it is mentioned in the context of neural recordings. I understand that PSTH is a common term in neuroscience that refers to a histogram showing the frequency of neural spikes (or responses) over time in response to a stimulus. It is typically calculated by averaging the spike trains across multiple trials for a given neuron. In the text, PSTH is used to represent the neural response that models are trying to predict (e.g., "the model predictions of the peri-stimulus time

histogram (PSTH, in grey) in normalized units (a.u.) as a function of time"). PSTH is obtained per neuron, as it reflects the response of a single neuron to a stimulus over time. So, how did you choose which neuron to show? What happens if you show another one? Is there a way to improve that example shown?

Minor comments:

The first time the acronym STRF appears (in the abstract) it is not defined, and that is not very clear for those of us who are not familiar with the model.

Review the grammar. Some examples I found are:

On the page "If STRFs can in this case be estimated using reverse correlation..." I think there is a grammatical error to correct.

"Gradmaps" → "GradMaps" (inconsistent capitalization).

"the PSTH were obtained" → "the PSTHs were obtained" (grammatical error, "PSTH" should be pluralized as "PSTHs").

Reviewer #2

(Remarks to the Author)

This study evaluates how well recurrent neural network models characterize natural sound representation in auditory cortex. Using large set of previously collected single-unit data, from several different labs and species, the authors fit models that incorporated different recurrent architectures and compared the prediction accuracy of these models to several previously established models. Across datasets, the recurrent models outperformed other models, including non-recurrent (stateless) deep learning-based models. The authors also develop a tool for visualizing the functional properties of these models by deriving the optimal stimulus predicted by the model.

The successful application of cutting-edge recurrent models to neurophysiological data represents an important advance in the use of machine learning tools to understand neural sensory processing. It is particularly noteworthy that the authors tested the new recurrent models using several datasets, and compared performance to several established model architectures. Together, these findings provide strong support for the utility of recurrent models to understand sensory neural data. However, there are some shortcomings that make it difficult to accept some of the findings, in particular, around whether the "Dreams" presented in the manuscript faithfully describe the functional properties that explain the improved performance of the recurrent models.

Main Concerns

1. (p. 4) "optimized Dream obtained with the GRU displays complex patterns even in the most ancient time steps..." How representative are the two examples? And is it possible to quantify how well the GradMap and/or Dream analyses tell us why the recurrent models are working? The two examples shown in the manuscript are quite interesting and complex (GRU and S4 models, Figs. 3, S3). But no population or validation analysis is performed to convince the reader that the result is capturing functional properties that matter for the model's performance.

A. It appears that the Dream measure is shown for just one unit. How normal is it for GRU, S4, or any model fits to have such long Dream stimuli? Some quantification should be provided for the entire dataset. It should be straightforward to measure some form of "memory" based on how far back in time the non-zero terms in the Dream extend. It may be that some of the stimulus sets have relatively short trials (unlike NAT4). Does this affect interpretation? It may make sense to focus the analysis on datasets with longer trials.

B. More important, can the authors argue that the very complex and specific Dream measure is a true descriptor of the neuron? Previous studies have validated optimal stimuli in closed-loop experiments, and it is understandably not possible for this dataset. Currently, the only argument supporting the conclusion is that long Dream sequences do not appear in the randomly initialized models. This does not seem like a very solid argument, as it could live in a space outside of tested stimuli and thus have no impact on model performance. There should be some way to test efficacy with cross validation. E.g., can the authors identify a stimulus that closely matches the Dream and show that the recurrent model is especially good at predicting the response compared to a stateless model? Some support is needed, or the results should be accompanied by a strong caveat that the Dreams have not been validated.

C. (p. 1, "arbitrarily chosen") Why are the above points (A, B) an important concern? The authors suggest in their introduction that work with stateless models has somewhat arbitrarily focused on encoding models with relatively short memory. But this is not the case. Much longer memory models have been tested, but these have shown that the longer integration times do not improve model prediction accuracy. Consider also recent work on integration times in human and ferret arguing for much shorter memory (Norman-Haignere et al 2022, Sabat et al bioRxiv 2025). There is some evidence for sparse, long-lasting contextual effects (Espejo & David 2024), which may not be captured by the approaches cited above. It would be exciting if the current study has identified these precise optimal patterns, but the claims are not consistent with most published evidence. Thus, some justification or caveat should be provided.

2. Related to 1A, the different datasets contain different amounts of data and trials of different duration. Does relative model performance depend on either of these aspects of the data? For example, trials in NAT4 were only 1 second long, so the recurrent model may be less beneficial for these stimuli. Some of the Wehr data only have 3 stimuli (according to Rancon et al 2024), and it seems that a more complex model may be difficult to fit with such limited data. At the very least, it would be helpful if the authors could provide a table comparing all the datasets analyzed, as in their previous publication. It would also be helpful to know if StateNet model performance depends on either dataset size or trial length.

Minor Concerns

General comment: There are a number of grammatical errors throughout the manuscript, and the authors might consider a careful proofread before resubmission. A couple have been flagged below.

(p. 1) “adaptive neural mechanisms” (first and second paragraphs) The language here is confusing. The authors’ list includes some actual neural mechanisms (SSA, STP) but also includes contrast gain control, “context dependence” and “priming” which are not really mechanisms, but rather functional phenomena. More careful language here would be helpful to help readers understand what phenomena the authors are trying to explain.

(p. 2) “unplausible” should be “implausible”

(p. 3) “Dream STRFs”. The authors might include a comment, either here or in the discussion, about previous observations that auditory neurons cannot be characterized by a single STRF (aka GradMap). Eg, Woolley et al 2006 showed that the STRF in songbird Field L differed if estimated with natural sounds vs. noise. Thus another way to characterize tuning might be to measure the STRF with different stimuli.

(p. 6) Fig. 2. This figure is quite complex, and the colors are a bit difficult to distinguish. Is there a better color palette, esp for LN/NRF/Dnet. One might consider putting the stateful models at the extreme right edge, indicating long memory, rather than at 0, suggesting short memory.

(p. 7) Can the authors quantify the similarity of the GradMaps to Dreams for each model architecture?

(p. 11) Typo: “mathematical details of the modules is ...” should be “are...”

(p. 14) I may have missed this somewhere, but how were model parameters initialized? Random values? Were multiple initial conditions tested?

(p. 14) Also regarding fitting, were any of the stateful architectures particularly more or less difficult to fit? One question that would be helpful to answer would be a suggestion, based on the authors’ experience, of what recurrent architecture is best-suited for neural data analysis. Nominally, GRU comes out on the top in Table 1, but maybe there are other factors to consider?

Version 1:

Reviewer comments:

Reviewer #1

(Remarks to the Author)

I have thoroughly reviewed this new manuscript version, as well as the authors’ responses, and I am satisfied. They have addressed all of my comments and suggestions thoughtfully. The revisions made are appropriate and improve the quality of the manuscript.

I believe the authors have done a great job responding to the feedback, and I consider the manuscript now ready for publication in its current form. The only additional comment I’d add is that the last figure has very similar colours for the different contributions, making it a bit difficult to distinguish. If they could fix this for the final version, it would be much better.

Reviewer #2

(Remarks to the Author)

The authors have very thoroughly addressed concerns raised in the previous review, and I have no substantive concerns about the revised submission.

Just a couple minor comments, which I will leave to the discretion of the editor:

The new manuscript is quite dense and in some places hard to follow. In particular I found the analysis of TBPTT to be quite complex. The authors might consider revising for clarity. It would also be helpful to always refer to time-related variables in

units of seconds, rather than time steps, which require the reader to translate to seconds in order to relate to the data.

One important point about the "Dreams" analysis is that the method is designed only to identify an optimal stimulus. One could imagine a number of contextual effects that might suppress responses, in a way that would not show up in an analysis trying to measure a maximal response. This might be a direction for future to mention in the discussion.

General remarks

We thank the two reviewers for their valuable feedback and constructive comments. Below is a point-by-point response to their comments. We provide with this resubmission marked up copies of the new versions of the manuscript and of the supplements.

Reviewer #1 (Remarks to the Author):

The paper introduces a new class of computational models known as "StateNet," which are based on deep recurrent neural networks (RNNs). These models demonstrate better performance than previous approaches in fitting neural responses within the auditory system. One key advantage of StateNet models is that they do not require predefined time windows to process auditory stimuli, unlike earlier stateless methods and transformer models. The authors' approach indicates that auditory neurons integrate information over much longer timescales—several seconds—than what current spectro-temporal receptive field (STRF) techniques suggest. I find the paper interesting; however, there are some details and comments that need to be addressed before publishing.

Regarding the introduction:

1) Could you give a definition, and a corresponding citation, of what you call xAI? It is a new term whose definition can be a bit vague.

Explainable AI (xAI) is a subfield of deep learning focused on dissecting the properties of trained neural networks to enhance their transparency and interpretability (see e.g., Adadi et al., 2018 or Gilpin et al., 2018). We have clarified this point in the new version of the manuscript.

New references cited:

Adadi, A., & Berrada, M. (2018). Peeking inside the black-box: a survey on explainable artificial intelligence (XAI). *IEEE access*, 6, 52138-52160.

Gilpin, L. H., Bau, D., Yuan, B. Z., Bajwa, A., Specter, M., & Kagal, L. (2018, October). Explaining explanations: An overview of interpretability of machine learning. In *2018 IEEE 5th International Conference on data science and advanced analytics (DSAA)* (pp. 80-89). IEEE.

Regarding the methods:

1) The text does not provide detailed information about the size (e.g., number of layers, neurons per layer) or specific characteristics of each neural network model (e.g., GRU, LSTM, Transformer). The work needs a table that compares which ones and how many parameters each of the neural networks (models) has. From reading the work, I do not understand how many input units the models have, nor how many units the hidden layer has according to the case. I understand that the output layer will have as many units as the corresponding data set. I would also like to know if the models that have fewer units at the output, corresponding to each data set, present a better accuracy or not, (to understand what happens with the dimensionality of the system).

Thank you for your feedback. In this revised version, we have included a table summarizing the model hyperparameters for each architecture and dataset (see Supplementary Table 3). It is important to note that two primary features may vary across datasets, significantly affecting the dimensionality of the models:

- the number of frequency bins in the stimulus spectrograms (F) has been maintained as provided in the original datasets. This choice ensures precise temporal alignment between the stimulus bins and responses, reduces the need for extensive preprocessing, and preserves consistency with the original publications.
- the number of output units (N), which is indeed equivalent to the number of cells recorded.

Consequently, we made slight adjustments to certain hyperparameters of the network architecture to ensure compatibility with each input-output format.

In order to address the second part of Reviewer #1's concern, we present below a plot illustrating the average test performance as a function of the number of output units. There is no significant correlation between these variables, as indicated by an R^2 value of 0.12. Given the negative outcome of this test, we do not think it is necessary to add this figure to the manuscript.

2) I would like you to better explain aspects of the training such as the temporal length of the spectrogram segments you use.

Each dataset was preprocessed to ensure that the stimulus-response clips used as training samples were maximized in length while retaining multiple trials within the recordings. This aspect is crucial during evaluation to accurately compute the CCnorm metric. Because of their different acquisition protocols, the sequence duration varied between datasets but also within some datasets: 5 s for NS1, 1.5 s for NAT4, 1-5 s for AA1, and up to 15 s for both Asari and Wehr datasets. These details are now provided in Supplementary Table 3. The intention was to enable stateful models to learn very long-range dependencies if needed. We have updated the “*Optimization process / model training*” subsection to clarify this point.

3) What happens if we analyze the dynamics from the point of view of the connectivity matrix and the activity produced by the artificial units, prior to the output layer, (the case of the vanilla RNN network). Can any structure be observed in the PCA component space (such as, for example, trajectories observed in the works that study motor control)? Can this work be compared with any of those works?

As detailed in the paper, our StateNet architecture projects the output prediction of neural activity at a given time step t as a linear function of the current hidden state components. Therefore, analyzing the hidden state dynamics can indeed provide insights into the properties of the neural responses that the model has learned during training.

We fully agree with Reviewer #1 that reducing the dimensionality of this latent space to 2 or 3 axes using techniques such as PCA would allow us to plot the trajectory during stimulus presentation. Our preliminary analyses actually included such representations. Below is a figure that we presented at the annual meeting of the Computational Neuroscience Society in Natal, Brazil, in 2024.

- *Panel A* displays the input spectrograms provided to a StateNet GRU model trained on the Asari Dataset. These spectrograms consist of two different "context" stimuli followed by the same "probe" stimulus.
- *Panel B* illustrates the responses to these two stimuli, showing both the actual recorded responses (top) and those predicted by the StateNet model and a trained 2D-CNN. While the 2D-CNN generates identical responses to the probe (dotted lines) due to its finite-duration STRF, the StateNet GRU model modulates the amplitude of the predicted oscillations (dashed) on the scale of seconds following the probe onset.
- Finally, *Panel C* displays the hidden state of the GRU model projected onto its three main principal components during the presentation of the probe. The oscillations in the predicted PSTH appear to be reflected here, but interpreting such a complex diagram remains challenging. Consequently, we have chosen to focus on our new tool based on gradient ascent (i.e., gradmaps and dreams) for interpreting our results

in the current version of the manuscript. We did not include these PCA analyses to maintain overall coherence in our analyses.

Results:

The first time the dataset appears, it does not have the corresponding citation to be able to easily locate the origin of the database.

Thanks. It was fixed in the new version.

It is not very clear to me how the results are shown in Figure 1. The concept of PSTH (Peri-Stimulus Time Histogram) is not explicitly defined in the text, but it is mentioned in the context of neural recordings. I understand that PSTH is a common term in neuroscience that refers to a histogram showing the frequency of neural spikes (or responses) over time in response to a stimulus. It is typically calculated by averaging the spike trains across multiple trials for a given neuron. In the text, PSTH is used to represent the neural response that models are trying to predict (e.g., "the model predictions of the peri-stimulus time histogram (PSTH, in grey) in normalized units (a.u.) as a function of time"). PSTH is obtained per neuron, as it reflects the response of a single neuron to a stimulus over time. So, how did you choose which neuron to show? What happens if you show another one? Is there a way to improve that example shown?

Thank you for your comment. PSTHs are indeed a well-established concept in neuroscience, representing the rate and timing of spike discharges in response to an external stimulus. We have now included an explicit definition in the "Materials and Methods" section (see page 15, "Neural Response Fitting Task", "Task definition"). Figure 1 illustrates examples of model predictions for various neural recordings. To offer a comprehensive overview, we employed different models, based on stateful (StateNet LSTM, GRU, and S4) and stateless (CNN, NRF, and Transformer) architectures. Additionally, we used neurons from diverse datasets (AA1, NAT4, and Wehr) and auditory areas (MLd, Field L, A1, and PEG). This clarification has been incorporated into the revised version of the manuscript. It is important to note that this figure serves solely as an illustration of the predicted neural responses for well-predicted neurons, allowing an overview of the recorded data and a better appreciation of what a high CC looks like for the latter. A detailed quantification of the model performances (CCnorm) across all datasets is presented in Table 1.

Minor comments:

The first time the acronym STRF appears (in the abstract) it is not defined, and that is not very clear for those of us who are not familiar with the model.

Thank you for reporting that. It was fixed in the new version.

Review the grammar. Some examples I found are:

On the page "If STRFs can in this case be estimated using reverse correlation..." I think there is a grammatical error to correct.

Thank you. It was fixed in the new version.

"Gradmaps" → "GradMaps" (inconsistent capitalization).

Thank you. It was fixed in the new version.

"the PSTH were obtained" → "the PSTHs were obtained" (grammatical error, "PSTH" should be pluralized as "PSTHs").

Thank you. It was fixed in the new version.

Reviewer #2 (Remarks to the Author):

This study evaluates how well recurrent neural network models characterize natural sound representation in auditory cortex. Using a large set of previously collected single-unit data, from several different labs and species, the authors fit models that incorporated different recurrent architectures and compared the prediction accuracy of these models to several previously established models. Across datasets, the recurrent models outperformed other models, including non-recurrent (stateless) deep learning-based models. The authors also develop a tool for visualizing the functional properties of these models by deriving the optimal stimulus predicted by the model.

The successful application of cutting-edge recurrent models to neurophysiological data represents an important advance in the use of machine learning tools to understand neural sensory processing. It is particularly noteworthy that the authors tested the new recurrent models using several datasets, and compared performance to several established model architectures. Together, these findings provide strong support for the utility of recurrent models to understand sensory neural data. However, there are some shortcomings that make it difficult to accept some of the findings, in particular, around whether the “Dreams” presented in the manuscript faithfully describe the functional properties that explain the improved performance of the recurrent models.

Main Concerns

1. (p. 4) “optimized Dream obtained with the GRU displays complex patterns even in the most ancient time steps...” How representative are the two examples? And is it possible to quantify how well the GradMap and/or Dream analyses tell us why the recurrent models are working? The two examples shown in the manuscript are quite interesting and complex (GRU and S4 models, Figs. 3, S3). But no population or validation analysis is performed to convince the reader that the result is capturing functional properties that matter for the model’s performance.

A. It appears that the Dream measure is shown for just one unit. How normal is it for GRU, S4, or any model fits to have such long Dream stimuli? Some quantification should be provided for the entire dataset. It should be straightforward to measure some form of “memory” based on how far back in time the non-zero terms in the Dream extend. It may be

that some of the stimulus sets have relatively short trials (unlike NAT4). Does this affect interpretation? It may make sense to focus the analysis on datasets with longer trials.

Thank you for these insightful comments. We fully concur with the reviewer on the importance of quantifying the contribution of past inputs to neural responses in our StateNet models. This concern is precisely why we developed the GradMap/Dream tool. Upon reflection, we realize that our previous submission did not adequately explain how to interpret these data, which may have led to some confusion. Specifically, for any given neural unit, the Dream tool generates the past stimulus that maximizes its current response. Dreams derived from the most elaborate StateNet models (e.g., GRU, S4) exhibit non-zero values at the earliest time steps, and this phenomenon is observed across many neurons. However, it is crucial to note that these non-zero values do not imply equal contributions to the neural response. For instance, in a Dream, two equivalent inputs for a given frequency but at different time-steps can have different impacts on the neural responses while both optimizing this response. We have clarified this point in the new version. Additionally, to quantify the relative contributions of past Dream values to neural responses in recurrent models, we computed Dreams of increasing duration (ranging from 0 ms to 1 s) for the 73 neural units in the NS1 dataset, and reported the elicited activations. The NS1 dataset was selected for its long sequences with high temporal resolution (featuring auditory inputs of 5 seconds divided into 1,000 time steps of 5 ms each) and high levels of predictability (nearly 80% CCnorm). The resulting curves for each StateNet model were averaged across the neural populations and normalized by their maximum values to facilitate comparison between models. The results of these analyses are presented in a new figure (Figure 6). Notably, except for the RNN model, where only Dream values within 100 ms of the neural responses contribute, we can observe that all other StateNet models (i.e., LSTM, GRU, S4, and Mamba) receive significant contributions from earlier time steps. However, these contributions are moderate, as all these models achieve at least 95% of their maximum responses using time windows of 500 ms (except for the S4 model, which only reaches 85% of its maximum).

To more precisely evaluate the relative contribution of each past instant (i.e., the memory of a model), we consider it more appropriate to use the initial GradMaps (G_0), which can be directly linked to the linear STRF. In this updated version, we calculated the GradMap energy as a function of latency for various classes of models (L/NL, NRF/DNet, 2D-CNN, Transformer, and StateNet) across all neurons in all datasets. The results of these analyses are presented in a new figure (Figure 5, panel b). These results indicate that StateNet models actually concentrate their energy in the latencies directly preceding the neural responses (i.e., the last 100 ms), similar to STRF models and consistent with recent studies suggested by this reviewer (Norman-Haignere et al., 2022; Sabat et al., 2025; see also our response to point C below). Furthermore, this figure confirms that GradMaps for StateNets are sparser compared to those for any other model (see also GradMap examples in Figure 2 and Figure S3), suggesting that this sparsity could be beneficial for performance accuracy.

In this new version, to provide further evidence that temporal recurrence is indeed the computational principle enabling state-of-the-art performance on the neural response fitting task, we have designed another experiment based on these GradMaps. This experiment involves training our StateNet architecture using a modified version of the Backpropagation Through Time (BPTT) algorithm. In this variant, known as Truncated

Backpropagation Through Time (TBPTT), the graph constructed by PyTorch's automatic differentiation engine to compute the gradient of the loss with respect to the model's parameters is truncated beyond a fixed number of past time steps. Consequently, a stateful model trained with this method can only learn shorter-term temporal dependencies, making the comparison with stateless models even more equitable. Despite this constraint, our StateNet architecture continues to outperform the best stateless approaches at both shorter and longer time scales, as presented in Fig. 5a. Moreover, training on long sequences with regular BPTT positively impacts the effective integration window, both quantitatively in terms of duration and qualitatively in terms of learned patterns. These results were performed across all neurons in the NS1 dataset and are shown in Figure 5, panels b and c.

We hope that these three new analyses offer a clearer overview of the properties of our StateNet models, particularly regarding their memory.

B. More important, can the authors argue that the very complex and specific Dream measure is a true descriptor of the neuron? Previous studies have validated optimal stimuli in closed-loop experiments, and it is understandably not possible for this dataset. Currently, the only argument supporting the conclusion is that long Dream sequences do not appear in the randomly initialized models. This does not seem like a very solid argument, as it could live in a space outside of tested stimuli and thus have no impact on model performance. There should be some way to test efficacy with cross validation. E.g., can the authors identify a stimulus that closely matches the Dream and show that the recurrent model is especially good at predicting the response compared to a stateless model? Some support is needed, or the results should be accompanied by a strong caveat that the Dreams have not been validated.

Thank you for these relevant remarks. In the previous version of the article, to ensure that the very long patterns visible in the Dreams of the more complex recurrent models were not just optimization artifacts, we compared them with randomly initialized models. We also conducted an experiment in which we compared the activation elicited by shorter dreams (bottom panels of Supplementary Figure S3). Random networks did not display such long patterns, suggesting that they have functional significance. Additionally, shorter dreams elicited weaker output activations. While the latter demonstrates that earlier time steps significantly contribute to neural activation, their contribution is weak and decreases in importance over time. We acknowledge that this experiment could have been further developed. Therefore, as discussed above, we have added a new figure (Figure 6) to quantify the contribution of each latency in the Dreams to the output activation, averaged across neurons.

We also considered searching for 'Dream-like' patterns in the dataset after training. However, we found that such an approach would suffer from circularity, as it would not be surprising to find audio patterns in the Dreams that were already present in the stimuli used to train the network. For this reason, we focused solely on test stimuli but did not find anything convincing to share due to the limited dataset size. Another challenge with the suggested approach is the long duration of the Dreams: while patterns present in test stimuli could potentially be found in the Dreams over the shorter latencies preceding neural activation, this is extremely unlikely over the longer latencies (~100-1000 ms) that make Dreams so intriguing. Indeed, in the unregularized formulation proposed in our paper, we

should not expect them to align perfectly with the distribution of natural stimuli over such durations, especially considering the minimal contribution of long latencies to the current response. Given that neurons can exhibit high responses to various stimuli and the model's strong fitting performance, Dreams *could* potentially drive neural activations effectively in an experimental setup. However, the sampling of the stimulus space during experimental data acquisition was very limited, making it likely that these stimuli were simply not presented to the animal. Thus, the key takeaway here is that new closed-loop experiments should be conducted to validate or invalidate our observations.

C. (p. 1, “arbitrarily chosen”) Why are the above points (A, B) an important concern? The authors suggest in their introduction that work with stateless models has somewhat arbitrarily focused on encoding models with relatively short memory. But this is not the case. Much longer memory models have been tested, but these have shown that the longer integration times do not improve model prediction accuracy. Consider also recent work on integration times in human and ferret arguing for much shorter memory (Norman-Haignere et al 2022, Sabat et al bioRxiv 2025). There is some evidence for sparse, long-lasting contextual effects (Espejo & David 2024), which may not be captured by the approaches cited above. It would be exciting if the current study has identified these precise optimal patterns, but the claims are not consistent with most published evidence. Thus, some justification or caveat should be provided.

Thank you very much for these highly relevant references, which we have now included in our manuscript; we have also updated this paragraph of the introduction to convey that integration times are actually selected based on experimental knowledge. Based on the additional analyses we conducted in response to the points raised by this reviewer (see above), we believe that our results align well with those reported in these two studies. This is because most of the energy in our gradmaps is contained within a relatively short time window (mostly < 250ms) before the neuron's response, as shown in the new Figure 4, panel a. However, the same analyses also indicated that earlier time windows still have a significant, albeit weak, impact on neural responses. We have clarified this point in the manuscript and hope that it now provides a better overview of our results.

2. Related to 1A, the different datasets contain different amounts of data and trials of different duration. Does relative model performance depend on either of these aspects of the data? For example, trials in NAT4 were only 1 second long, so the recurrent model may be less beneficial for these stimuli. Some of the Wehr data only have 3 stimuli (according to Rancon et al 2024), and it seems that a more complex model may be difficult to fit with such limited data. At the very least, it would be helpful if the authors could provide a table comparing all the datasets analyzed, as in their previous publication. It would also be helpful to know if StateNet model performance depends on either dataset size or trial length.

Thank you for this comment, which has also been raised by Reviewer #1 (see their first point regarding the methods). We now provide the main features of each dataset, as well as the hyperparameters used in the associated models, in Supplementary Figure 3. Additionally, we investigated a potential correlation between StateNet performance and dataset size (in terms of the number of stimulus-response sequences per neuron) but did not observe any particular pattern ($R^2 = 0.06$), as illustrated in the Figure below. Given the

negative outcome of this test, we do not believe it is necessary to add this figure to the manuscript.

Finally, our experiments with TBPTT (see above) offer a comprehensive answer as to the relationship between stimulus duration and StateNet performance.

Minor Concerns

General comment: There are a number of grammatical errors throughout the manuscript, and the authors might consider a careful proofread before resubmission. A couple have been flagged below.

(p. 1) “adaptive neural mechanisms” (first and second paragraphs) The language here is confusing. The authors’ list includes some actual neural mechanisms (SSA, STP) but also includes contrast gain control, “context dependence” and “priming” which are not really mechanisms, but rather functional phenomena. More careful language here would be helpful to help readers understand what phenomena the authors are trying to explain.

Thank you for this comment. We have tried to clarify these points in the new version of the manuscript.

(p. 2) “unplausible” should be “implausible”

Thank you. It was fixed in the new version.

(p. 3) “Dream STRFs”. The authors might include a comment, either here or in the discussion, about previous observations that auditory neurons cannot be characterized by a single STRF (aka GradMap). Eg, Woolley et al 2006 showed that the STRF in songbird Field

L differed if estimated with natural sounds vs. noise. Thus another way to characterize tuning might be to measure the STRF with different stimuli.

Thank you for this comment. It has indeed been extensively shown in the literature that STRFs depend on the stimuli used to derive them (Theunissen et al., 2000; Woolley et al., 2006). Please note that, much like traditional STRFs, our models' weights and biases (and therefore their GradMaps and Dreams) also depend on the stimuli used to train them. This property is inherited from the spike-triggered average (STA), the electrophysiology technique that GradMaps is designed to mimic. However, in this study, we have focused on natural stimuli, as they generally yield higher explanatory power than artificial stimuli (such as white noise, TORCs, etc.) in terms of better generalization to other stimulus classes (Theunissen et al., 2000; Sharpee et al., 2008; Wang et al., 2025). In future work, we plan to further characterize GradMaps, explore their relationship with the Linear STRF derived from STA, and extend this method. For example, one possibility that this tool offers is the offline calculation of GradMaps following a context sound (e.g., a conspecific vocalization). Comparing this with the GradMap obtained without context should enable us to better understand how the context stimulus modulates the neuron's input-output function. We have added these elements and references in the discussion.

(p. 6) Fig. 2. This figure is quite complex, and the colors are a bit difficult to distinguish. Is there a better color palette, esp for LN/NRF/Dnet. One might consider putting the stateful models at the extreme right edge, indicating long memory, rather than at 0, suggesting short memory.

Thank you for your comment. We apologize for any readability issues, and have therefore increased the size of the figure, which only occupied half the space of the page despite its complexity. We have also updated the color palette of the above models in all our figures to enhance readability and to remain consistent with our conventions (shades of the same color for models of the same family). Regarding the positioning of StateNet models in these graphs, we chose to place them on the left-hand side (i.e., short *explicit* memory) to adhere to plotting conventions found in the literature (Rahman et al., 2019, "A dynamic network model of temporal receptive fields in primary auditory cortex," PLoS Computational Biology) and to maintain coherence with our argument about delay lines and the scaling of model size with Temporal Receptive Field (TRF) size. Figure 5a now presents in detail the scaling of performances of a StateNet model as a function of its *implicit* memory, with long memory on the right-hand side, as suggested.

(p. 7) Can the authors quantify the similarity of the GradMaps to Dreams for each model architecture?

We now present an additional analysis that further characterizes models at the population level using GradMaps. By computing the correlation between these GradMaps across different neurons for various models, we uncovered distinct clusters. These clusters reflect the "computational family" to which the models belong, as illustrated in the similarity matrix now shown in Fig. 4.

(p. 11) Typo: "mathematical details of the modules is ..." should be "are..."

Thank you. It was fixed in the new version.

(p. 14) I may have missed this somewhere, but how were model parameters initialized? Random values? Were multiple initial conditions tested?

Model parameters were initialized with random values. We have clarified this point in the “*Optimization process / model training*” subsection.

(p. 14) Also regarding fitting, were any of the stateful architectures particularly more or less difficult to fit? One question that would be helpful to answer would be a suggestion, based on the authors’ experience, of what recurrent architecture is best-suited for neural data analysis. Nominally, GRU comes out on the top in Table 1, but maybe there are other factors to consider?

All models were relatively straightforward to fit, as we did not find any specific factors (such as initialization, layer configuration, or weight decay) crucial for the convergence of the training process. Among stateful models, we recommend using the GRU due to its performance, simplicity in formalism (requiring one fewer equation than the LSTM), and a lower number of parameters. However, given the extensive range of mathematical tools available for analyzing SSM models, they could also be a favorable choice for potentially enhanced interpretability while maintaining high performance levels. We have now added a brief discussion on this topic in the discussion.

General remarks

We thank the two reviewers for their valuable feedback and constructive comments. Below is a point-by-point response to their comments. We provide with this resubmission marked up copies of the new versions of the manuscript and of the supplements.

Reviewer #1 (Remarks to the Author):

The paper introduces a new class of computational models known as "StateNet," which are based on deep recurrent neural networks (RNNs). These models demonstrate better performance than previous approaches in fitting neural responses within the auditory system. One key advantage of StateNet models is that they do not require predefined time windows to process auditory stimuli, unlike earlier stateless methods and transformer models. The authors' approach indicates that auditory neurons integrate information over much longer timescales—several seconds—than what current spectro-temporal receptive field (STRF) techniques suggest. I find the paper interesting; however, there are some details and comments that need to be addressed before publishing.

Regarding the introduction:

1) Could you give a definition, and a corresponding citation, of what you call xAI? It is a new term whose definition can be a bit vague.

Explainable AI (xAI) is a subfield of deep learning focused on dissecting the properties of trained neural networks to enhance their transparency and interpretability (see e.g., Adadi et al., 2018 or Gilpin et al., 2018). We have clarified this point in the new version of the manuscript.

New references cited:

Adadi, A., & Berrada, M. (2018). Peeking inside the black-box: a survey on explainable artificial intelligence (XAI). *IEEE access*, 6, 52138-52160.

Gilpin, L. H., Bau, D., Yuan, B. Z., Bajwa, A., Specter, M., & Kagal, L. (2018, October). Explaining explanations: An overview of interpretability of machine learning. In *2018 IEEE 5th International Conference on data science and advanced analytics (DSAA)* (pp. 80-89). IEEE.

Regarding the methods:

1) The text does not provide detailed information about the size (e.g., number of layers, neurons per layer) or specific characteristics of each neural network model (e.g., GRU, LSTM, Transformer). The work needs a table that compares which ones and how many parameters each of the neural networks (models) has. From reading the work, I do not understand how many input units the models have, nor how many units the hidden layer has according to the case. I understand that the output layer will have as many units as the corresponding data set. I would also like to know if the models that have fewer units at the output, corresponding to each data set, present a better accuracy or not, (to understand what happens with the dimensionality of the system).

Thank you for your feedback. In this revised version, we have included a table summarizing the model hyperparameters for each architecture and dataset (see Supplementary Table 3). It is important to note that two primary features may vary across datasets, significantly affecting the dimensionality of the models:

- the number of frequency bins in the stimulus spectrograms (F) has been maintained as provided in the original datasets. This choice ensures precise temporal alignment between the stimulus bins and responses, reduces the need for extensive preprocessing, and preserves consistency with the original publications.
- the number of output units (N), which is indeed equivalent to the number of cells recorded.

Consequently, we made slight adjustments to certain hyperparameters of the network architecture to ensure compatibility with each input-output format.

In order to address the second part of Reviewer #1's concern, we present below a plot illustrating the average test performance as a function of the number of output units. There is no significant correlation between these variables, as indicated by an R^2 value of 0.12. Given the negative outcome of this test, we do not think it is necessary to add this figure to the manuscript.

2) I would like you to better explain aspects of the training such as the temporal length of the spectrogram segments you use.

Each dataset was preprocessed to ensure that the stimulus-response clips used as training samples were maximized in length while retaining multiple trials within the recordings. This aspect is crucial during evaluation to accurately compute the CCnorm metric. Because of their different acquisition protocols, the sequence duration varied between datasets but also within some datasets: 5 s for NS1, 1.5 s for NAT4, 1-5 s for AA1, and up to 15 s for both Asari and Wehr datasets. These details are now provided in Supplementary Table 3. The intention was to enable stateful models to learn very long-range dependencies if needed. We have updated the “*Optimization process / model training*” subsection to clarify this point.

3) What happens if we analyze the dynamics from the point of view of the connectivity matrix and the activity produced by the artificial units, prior to the output layer, (the case of the vanilla RNN network). Can any structure be observed in the PCA component space (such as, for example, trajectories observed in the works that study motor control)? Can this work be compared with any of those works?

As detailed in the paper, our StateNet architecture projects the output prediction of neural activity at a given time step t as a linear function of the current hidden state components. Therefore, analyzing the hidden state dynamics can indeed provide insights into the properties of the neural responses that the model has learned during training.

We fully agree with Reviewer #1 that reducing the dimensionality of this latent space to 2 or 3 axes using techniques such as PCA would allow us to plot the trajectory during stimulus presentation. Our preliminary analyses actually included such representations. Below is a figure that we presented at the annual meeting of the Computational Neuroscience Society in Natal, Brazil, in 2024.

- *Panel A* displays the input spectrograms provided to a StateNet GRU model trained on the Asari Dataset. These spectrograms consist of two different "context" stimuli followed by the same "probe" stimulus.
- *Panel B* illustrates the responses to these two stimuli, showing both the actual recorded responses (top) and those predicted by the StateNet model and a trained 2D-CNN. While the 2D-CNN generates identical responses to the probe (dotted lines) due to its finite-duration STRF, the StateNet GRU model modulates the amplitude of the predicted oscillations (dashed) on the scale of seconds following the probe onset.
- Finally, *Panel C* displays the hidden state of the GRU model projected onto its three main principal components during the presentation of the probe. The oscillations in the predicted PSTH appear to be reflected here, but interpreting such a complex diagram remains challenging. Consequently, we have chosen to focus on our new tool based on gradient ascent (i.e., gradmaps and dreams) for interpreting our results

in the current version of the manuscript. We did not include these PCA analyses to maintain overall coherence in our analyses.

Results:

The first time the dataset appears, it does not have the corresponding citation to be able to easily locate the origin of the database.

Thanks. It was fixed in the new version.

It is not very clear to me how the results are shown in Figure 1. The concept of PSTH (Peri-Stimulus Time Histogram) is not explicitly defined in the text, but it is mentioned in the context of neural recordings. I understand that PSTH is a common term in neuroscience that refers to a histogram showing the frequency of neural spikes (or responses) over time in response to a stimulus. It is typically calculated by averaging the spike trains across multiple trials for a given neuron. In the text, PSTH is used to represent the neural response that models are trying to predict (e.g., "the model predictions of the peri-stimulus time histogram (PSTH, in grey) in normalized units (a.u.) as a function of time"). PSTH is obtained per neuron, as it reflects the response of a single neuron to a stimulus over time. So, how did you choose which neuron to show? What happens if you show another one? Is there a way to improve that example shown?

Thank you for your comment. PSTHs are indeed a well-established concept in neuroscience, representing the rate and timing of spike discharges in response to an external stimulus. We have now included an explicit definition in the "Materials and Methods" section (see page 15, "Neural Response Fitting Task", "Task definition"). Figure 1 illustrates examples of model predictions for various neural recordings. To offer a comprehensive overview, we employed different models, based on stateful (StateNet LSTM, GRU, and S4) and stateless (CNN, NRF, and Transformer) architectures. Additionally, we used neurons from diverse datasets (AA1, NAT4, and Wehr) and auditory areas (MLd, Field L, A1, and PEG). This clarification has been incorporated into the revised version of the manuscript. It is important to note that this figure serves solely as an illustration of the predicted neural responses for well-predicted neurons, allowing an overview of the recorded data and a better appreciation of what a high CC looks like for the latter. A detailed quantification of the model performances (CCnorm) across all datasets is presented in Table 1.

Minor comments:

The first time the acronym STRF appears (in the abstract) it is not defined, and that is not very clear for those of us who are not familiar with the model.

Thank you for reporting that. It was fixed in the new version.

Review the grammar. Some examples I found are:

On the page "If STRFs can in this case be estimated using reverse correlation..." I think there is a grammatical error to correct.

Thank you. It was fixed in the new version.

"Gradmaps" → "GradMaps" (inconsistent capitalization).

Thank you. It was fixed in the new version.

"the PSTH were obtained" → "the PSTHs were obtained" (grammatical error, "PSTH" should be pluralized as "PSTHs").

Thank you. It was fixed in the new version.

Reviewer #2 (Remarks to the Author):

This study evaluates how well recurrent neural network models characterize natural sound representation in auditory cortex. Using a large set of previously collected single-unit data, from several different labs and species, the authors fit models that incorporated different recurrent architectures and compared the prediction accuracy of these models to several previously established models. Across datasets, the recurrent models outperformed other models, including non-recurrent (stateless) deep learning-based models. The authors also develop a tool for visualizing the functional properties of these models by deriving the optimal stimulus predicted by the model.

The successful application of cutting-edge recurrent models to neurophysiological data represents an important advance in the use of machine learning tools to understand neural sensory processing. It is particularly noteworthy that the authors tested the new recurrent models using several datasets, and compared performance to several established model architectures. Together, these findings provide strong support for the utility of recurrent models to understand sensory neural data. However, there are some shortcomings that make it difficult to accept some of the findings, in particular, around whether the “Dreams” presented in the manuscript faithfully describe the functional properties that explain the improved performance of the recurrent models.

Main Concerns

1. (p. 4) “optimized Dream obtained with the GRU displays complex patterns even in the most ancient time steps...” How representative are the two examples? And is it possible to quantify how well the GradMap and/or Dream analyses tell us why the recurrent models are working? The two examples shown in the manuscript are quite interesting and complex (GRU and S4 models, Figs. 3, S3). But no population or validation analysis is performed to convince the reader that the result is capturing functional properties that matter for the model’s performance.

A. It appears that the Dream measure is shown for just one unit. How normal is it for GRU, S4, or any model fits to have such long Dream stimuli? Some quantification should be provided for the entire dataset. It should be straightforward to measure some form of “memory” based on how far back in time the non-zero terms in the Dream extend. It may be

that some of the stimulus sets have relatively short trials (unlike NAT4). Does this affect interpretation? It may make sense to focus the analysis on datasets with longer trials.

Thank you for these insightful comments. We fully concur with the reviewer on the importance of quantifying the contribution of past inputs to neural responses in our StateNet models. This concern is precisely why we developed the GradMap/Dream tool. Upon reflection, we realize that our previous submission did not adequately explain how to interpret these data, which may have led to some confusion. Specifically, for any given neural unit, the Dream tool generates the past stimulus that maximizes its current response. Dreams derived from the most elaborate StateNet models (e.g., GRU, S4) exhibit non-zero values at the earliest time steps, and this phenomenon is observed across many neurons. However, it is crucial to note that these non-zero values do not imply equal contributions to the neural response. For instance, in a Dream, two equivalent inputs for a given frequency but at different time-steps can have different impacts on the neural responses while both optimizing this response. We have clarified this point in the new version. Additionally, to quantify the relative contributions of past Dream values to neural responses in recurrent models, we computed Dreams of increasing duration (ranging from 0 ms to 1 s) for the 73 neural units in the NS1 dataset, and reported the elicited activations. The NS1 dataset was selected for its long sequences with high temporal resolution (featuring auditory inputs of 5 seconds divided into 1,000 time steps of 5 ms each) and high levels of predictability (nearly 80% CCnorm). The resulting curves for each StateNet model were averaged across the neural populations and normalized by their maximum values to facilitate comparison between models. The results of these analyses are presented in a new figure (Figure 6). Notably, except for the RNN model, where only Dream values within 100 ms of the neural responses contribute, we can observe that all other StateNet models (i.e., LSTM, GRU, S4, and Mamba) receive significant contributions from earlier time steps. However, these contributions are moderate, as all these models achieve at least 95% of their maximum responses using time windows of 500 ms (except for the S4 model, which only reaches 85% of its maximum).

To more precisely evaluate the relative contribution of each past instant (i.e., the memory of a model), we consider it more appropriate to use the initial GradMaps (G_0), which can be directly linked to the linear STRF. In this updated version, we calculated the GradMap energy as a function of latency for various classes of models (L/NL, NRF/DNet, 2D-CNN, Transformer, and StateNet) across all neurons in all datasets. The results of these analyses are presented in a new figure (Figure 5, panel b). These results indicate that StateNet models actually concentrate their energy in the latencies directly preceding the neural responses (i.e., the last 100 ms), similar to STRF models and consistent with recent studies suggested by this reviewer (Norman-Haignere et al., 2022; Sabat et al., 2025; see also our response to point C below). Furthermore, this figure confirms that GradMaps for StateNets are sparser compared to those for any other model (see also GradMap examples in Figure 2 and Figure S3), suggesting that this sparsity could be beneficial for performance accuracy.

In this new version, to provide further evidence that temporal recurrence is indeed the computational principle enabling state-of-the-art performance on the neural response fitting task, we have designed another experiment based on these GradMaps. This experiment involves training our StateNet architecture using a modified version of the Backpropagation Through Time (BPTT) algorithm. In this variant, known as Truncated

Backpropagation Through Time (TBPTT), the graph constructed by PyTorch's automatic differentiation engine to compute the gradient of the loss with respect to the model's parameters is truncated beyond a fixed number of past time steps. Consequently, a stateful model trained with this method can only learn shorter-term temporal dependencies, making the comparison with stateless models even more equitable. Despite this constraint, our StateNet architecture continues to outperform the best stateless approaches at both shorter and longer time scales, as presented in Fig. 5a. Moreover, training on long sequences with regular BPTT positively impacts the effective integration window, both quantitatively in terms of duration and qualitatively in terms of learned patterns. These results were performed across all neurons in the NS1 dataset and are shown in Figure 5, panels b and c.

We hope that these three new analyses offer a clearer overview of the properties of our StateNet models, particularly regarding their memory.

B. More important, can the authors argue that the very complex and specific Dream measure is a true descriptor of the neuron? Previous studies have validated optimal stimuli in closed-loop experiments, and it is understandably not possible for this dataset. Currently, the only argument supporting the conclusion is that long Dream sequences do not appear in the randomly initialized models. This does not seem like a very solid argument, as it could live in a space outside of tested stimuli and thus have no impact on model performance. There should be some way to test efficacy with cross validation. E.g., can the authors identify a stimulus that closely matches the Dream and show that the recurrent model is especially good at predicting the response compared to a stateless model? Some support is needed, or the results should be accompanied by a strong caveat that the Dreams have not been validated.

Thank you for these relevant remarks. In the previous version of the article, to ensure that the very long patterns visible in the Dreams of the more complex recurrent models were not just optimization artifacts, we compared them with randomly initialized models. We also conducted an experiment in which we compared the activation elicited by shorter dreams (bottom panels of Supplementary Figure S3). Random networks did not display such long patterns, suggesting that they have functional significance. Additionally, shorter dreams elicited weaker output activations. While the latter demonstrates that earlier time steps significantly contribute to neural activation, their contribution is weak and decreases in importance over time. We acknowledge that this experiment could have been further developed. Therefore, as discussed above, we have added a new figure (Figure 6) to quantify the contribution of each latency in the Dreams to the output activation, averaged across neurons.

We also considered searching for 'Dream-like' patterns in the dataset after training. However, we found that such an approach would suffer from circularity, as it would not be surprising to find audio patterns in the Dreams that were already present in the stimuli used to train the network. For this reason, we focused solely on test stimuli but did not find anything convincing to share due to the limited dataset size. Another challenge with the suggested approach is the long duration of the Dreams: while patterns present in test stimuli could potentially be found in the Dreams over the shorter latencies preceding neural activation, this is extremely unlikely over the longer latencies (~100-1000 ms) that make Dreams so intriguing. Indeed, in the unregularized formulation proposed in our paper, we

should not expect them to align perfectly with the distribution of natural stimuli over such durations, especially considering the minimal contribution of long latencies to the current response. Given that neurons can exhibit high responses to various stimuli and the model's strong fitting performance, Dreams *could* potentially drive neural activations effectively in an experimental setup. However, the sampling of the stimulus space during experimental data acquisition was very limited, making it likely that these stimuli were simply not presented to the animal. Thus, the key takeaway here is that new closed-loop experiments should be conducted to validate or invalidate our observations.

C. (p. 1, “arbitrarily chosen”) Why are the above points (A, B) an important concern? The authors suggest in their introduction that work with stateless models has somewhat arbitrarily focused on encoding models with relatively short memory. But this is not the case. Much longer memory models have been tested, but these have shown that the longer integration times do not improve model prediction accuracy. Consider also recent work on integration times in human and ferret arguing for much shorter memory (Norman-Haignere et al 2022, Sabat et al bioRxiv 2025). There is some evidence for sparse, long-lasting contextual effects (Espejo & David 2024), which may not be captured by the approaches cited above. It would be exciting if the current study has identified these precise optimal patterns, but the claims are not consistent with most published evidence. Thus, some justification or caveat should be provided.

Thank you very much for these highly relevant references, which we have now included in our manuscript; we have also updated this paragraph of the introduction to convey that integration times are actually selected based on experimental knowledge. Based on the additional analyses we conducted in response to the points raised by this reviewer (see above), we believe that our results align well with those reported in these two studies. This is because most of the energy in our gradmaps is contained within a relatively short time window (mostly < 250ms) before the neuron's response, as shown in the new Figure 4, panel a. However, the same analyses also indicated that earlier time windows still have a significant, albeit weak, impact on neural responses. We have clarified this point in the manuscript and hope that it now provides a better overview of our results.

2. Related to 1A, the different datasets contain different amounts of data and trials of different duration. Does relative model performance depend on either of these aspects of the data? For example, trials in NAT4 were only 1 second long, so the recurrent model may be less beneficial for these stimuli. Some of the Wehr data only have 3 stimuli (according to Rancon et al 2024), and it seems that a more complex model may be difficult to fit with such limited data. At the very least, it would be helpful if the authors could provide a table comparing all the datasets analyzed, as in their previous publication. It would also be helpful to know if StateNet model performance depends on either dataset size or trial length.

Thank you for this comment, which has also been raised by Reviewer #1 (see their first point regarding the methods). We now provide the main features of each dataset, as well as the hyperparameters used in the associated models, in Supplementary Figure 3. Additionally, we investigated a potential correlation between StateNet performance and dataset size (in terms of the number of stimulus-response sequences per neuron) but did not observe any particular pattern ($R^2 = 0.06$), as illustrated in the Figure below. Given the

negative outcome of this test, we do not believe it is necessary to add this figure to the manuscript.

Finally, our experiments with TBPTT (see above) offer a comprehensive answer as to the relationship between stimulus duration and StateNet performance.

Minor Concerns

General comment: There are a number of grammatical errors throughout the manuscript, and the authors might consider a careful proofread before resubmission. A couple have been flagged below.

(p. 1) “adaptive neural mechanisms” (first and second paragraphs) The language here is confusing. The authors’ list includes some actual neural mechanisms (SSA, STP) but also includes contrast gain control, “context dependence” and “priming” which are not really mechanisms, but rather functional phenomena. More careful language here would be helpful to help readers understand what phenomena the authors are trying to explain.

Thank you for this comment. We have tried to clarify these points in the new version of the manuscript.

(p. 2) “unplausible” should be “implausible”

Thank you. It was fixed in the new version.

(p. 3) “Dream STRFs”. The authors might include a comment, either here or in the discussion, about previous observations that auditory neurons cannot be characterized by a single STRF (aka GradMap). Eg, Woolley et al 2006 showed that the STRF in songbird Field

L differed if estimated with natural sounds vs. noise. Thus another way to characterize tuning might be to measure the STRF with different stimuli.

Thank you for this comment. It has indeed been extensively shown in the literature that STRFs depend on the stimuli used to derive them (Theunissen et al., 2000; Woolley et al., 2006). Please note that, much like traditional STRFs, our models' weights and biases (and therefore their GradMaps and Dreams) also depend on the stimuli used to train them. This property is inherited from the spike-triggered average (STA), the electrophysiology technique that GradMaps is designed to mimic. However, in this study, we have focused on natural stimuli, as they generally yield higher explanatory power than artificial stimuli (such as white noise, TORCs, etc.) in terms of better generalization to other stimulus classes (Theunissen et al., 2000; Sharpee et al., 2008; Wang et al., 2025). In future work, we plan to further characterize GradMaps, explore their relationship with the Linear STRF derived from STA, and extend this method. For example, one possibility that this tool offers is the offline calculation of GradMaps following a context sound (e.g., a conspecific vocalization). Comparing this with the GradMap obtained without context should enable us to better understand how the context stimulus modulates the neuron's input-output function. We have added these elements and references in the discussion.

(p. 6) Fig. 2. This figure is quite complex, and the colors are a bit difficult to distinguish. Is there a better color palette, esp for LN/NRF/Dnet. One might consider putting the stateful models at the extreme right edge, indicating long memory, rather than at 0, suggesting short memory.

Thank you for your comment. We apologize for any readability issues, and have therefore increased the size of the figure, which only occupied half the space of the page despite its complexity. We have also updated the color palette of the above models in all our figures to enhance readability and to remain consistent with our conventions (shades of the same color for models of the same family). Regarding the positioning of StateNet models in these graphs, we chose to place them on the left-hand side (i.e., short *explicit* memory) to adhere to plotting conventions found in the literature (Rahman et al., 2019, "A dynamic network model of temporal receptive fields in primary auditory cortex," PLoS Computational Biology) and to maintain coherence with our argument about delay lines and the scaling of model size with Temporal Receptive Field (TRF) size. Figure 5a now presents in detail the scaling of performances of a StateNet model as a function of its *implicit* memory, with long memory on the right-hand side, as suggested.

(p. 7) Can the authors quantify the similarity of the GradMaps to Dreams for each model architecture?

We now present an additional analysis that further characterizes models at the population level using GradMaps. By computing the correlation between these GradMaps across different neurons for various models, we uncovered distinct clusters. These clusters reflect the "computational family" to which the models belong, as illustrated in the similarity matrix now shown in Fig. 4.

(p. 11) Typo: "mathematical details of the modules is ..." should be "are..."

Thank you. It was fixed in the new version.

(p. 14) I may have missed this somewhere, but how were model parameters initialized? Random values? Were multiple initial conditions tested?

Model parameters were initialized with random values. We have clarified this point in the “*Optimization process / model training*” subsection.

(p. 14) Also regarding fitting, were any of the stateful architectures particularly more or less difficult to fit? One question that would be helpful to answer would be a suggestion, based on the authors’ experience, of what recurrent architecture is best-suited for neural data analysis. Nominally, GRU comes out on the top in Table 1, but maybe there are other factors to consider?

All models were relatively straightforward to fit, as we did not find any specific factors (such as initialization, layer configuration, or weight decay) crucial for the convergence of the training process. Among stateful models, we recommend using the GRU due to its performance, simplicity in formalism (requiring one fewer equation than the LSTM), and a lower number of parameters. However, given the extensive range of mathematical tools available for analyzing SSM models, they could also be a favorable choice for potentially enhanced interpretability while maintaining high performance levels. We have now added a brief discussion on this topic in the discussion.

General remarks

We would like to thank both two reviewers again for their valuable feedback and constructive comments. Below is a point-by-point response to their remaining comments. We provide with this revision marked up copies of the new versions of the article.

Reviewer #1 (Remarks to the Author):

I have thoroughly reviewed this new manuscript version, as well as the authors' responses, and I am satisfied. They have addressed all of my comments and suggestions thoughtfully. The revisions made are appropriate and improve the quality of the manuscript. I believe the authors have done a great job responding to the feedback, and I consider the manuscript now ready for publication in its current form. The only additional comment I'd add is that the last figure has very similar colours for the different contributions, making it a bit difficult to distinguish. If they could fix this for the final version, it would be much better.

Thank you for your feedback. We recognize that the previous version of this figure was difficult to interpret. To ensure consistency with the manuscript's graphic style, we have revised it and now use distinct markers for each model/curve.

Reviewer #2 (Remarks to the Author):

The authors have very thoroughly addressed concerns raised in the previous review, and I have no substantive concerns about the revised submission.

Just a couple minor comments, which I will leave to the discretion of the editor:

The new manuscript is quite dense and in some places hard to follow. In particular I found the analysis of TBPTT to be quite complex. The authors might consider revising for clarity.

Thank you for your comment. In this new version, we edited the corresponding paragraph of the Results section ("*Truncated Back-Propagation Through Time reveals the long-term memory of recurrent models*") and Materials and Methods to improve clarity.

It would also be helpful to always refer to time-related variables in units of seconds, rather than time steps, which require the reader to translate to seconds in order to relate to the data.

Thank you for your comment. In this revised version, we have converted the time axis of all the figures to milliseconds. However, please note that this conversion was not possible for the curves in Fig. 4b, as they were generated by averaging datasets with different temporal resolutions. Therefore, the time axis for this figure remains in time steps.

One important point about the "Dreams" analysis is that the method is designed only to identify an optimal stimulus. One could imagine a number of contextual effects that might suppress responses, in a way that would not show up in an analysis trying to measure a maximal response. This might be a direction for the future to mention in the discussion.

Thank you for this remark. To address this concern, we edited a part of the Discussion in which we mentioned contextual effects.